# Effects of oxidative stress on hepatic encephalopathy pathogenesis in mice

Yunhu Bai[1,2,5], Kenan Li[3,5], Xiaodong Li[1], Xiyu Chen[1], Jie Zheng[3], Feifei Wu[3], Jinghao Chen[1], Ze Li[3], Shuai Zhang[3], Kun Wu[4], Yong Chen[1] ✉, Yayun Wang [3] ✉ & Yanling Yang[1] ✉

Oxidative stress plays a crucial role in the pathogenesis of hepatic encephalopathy (HE), but the mechanism remains unclear. GABAergic neurons in substantia nigra pars reticulata (SNr) contribute to the motor deficit of HE. The present study aims to investigate the effects of oxidative stress on HE in male mice. The results validate the existence of oxidative stress in both liver and SNr across two murine models of HE induced by thioacetamide (TAA) and bile duct ligation (BDL). Systemic mitochondria-targeted antioxidative drug mitoquinone (Mito-Q) rescues mitochondrial dysfunction and oxidative injury in SNr, so as to restore the locomotor impairment in TAA and BDL mice. Furthermore, the GAD2-expressing SNr population (SNr^GAD2) is activated by HE. Both overexpression of mitochondrial uncoupling protein 2 (UCP2) targeted to SNr^GAD2 and SNr^GAD2-targeted chemogenetic inhibition targeted to SNr^GAD2 rescue mitochondrial dysfunction in TAA-induced HE. These results define the key role of oxidative stress in the pathogenesis of HE.

Hepatic encephalopathy (HE) refers to the disruptions in the brain function resulting from acute or chronic liver failure or portal shunt[1,2]. The increase in hospitalization rates imposes a heavy burden on health care systems[3]. One of the early symptoms of HE is bradykinesia, which is characterized by the slow locomotor activity and impaired motor coordination[4]. It is important to study the underlying mechanism of bradykinesia in HE.

A growing number of studies have indicated that ammonia could induce the formation of reactive oxygen species (ROS) and increase oxidative stress levels[5,6]. It has been found that ROS could trigger a series of adverse effects, especially on mitochondrial morphology and function as well as neuronal activity[7,8]. Significantly, recent clinical reports have confirmed that the early reduction of oxidative stress in HE patients could minimize the occurrence of severe neurotoxicity. Oxidative stress is of great importance in the pathogenesis of HE, while the underlying mechanism remains to be clarified.

The motor activity is modulated by GABA and glutamate in basal ganglia–thalamus–cortex circuit[9]. γ-aminobutyric acid (GABA) is the principal inhibitory neurotransmitter. GABA-releasing (GABAergic) neurons, play a vital role as inhibitory neurons of the central nervous system in mammals as well as the human brain. In 1982, Schafer and Jones have proposed the hypothesis that the increased "GABAergic tone" contributes to the pathogenesis of HE. Since then, a growing number of evidence has shown that HE induces an increase in GABA level in various parts of the brain, including the cerebellum and the hippocampus[10,11]. Additionally, it has been revealed that the GABA_A receptor levels increase in the brain of HE animals[12,13]. A recent article in *Science* reports that the optogenetic activation of glutamic acid decarboxylase 2 (GAD2) neurons in the SNr leads to the significant enhancement of movement termination[14]. Nevertheless, it remains unclear whether HE can be activated by either itself or local high ammonia levels that directly stimulate GABAergic neurons. Additionally, the impact on HE from chemogenetic manipulating, these types of

[1]Department of Hepatobiliary Surgery, Xi-Jing Hospital, The Fourth Military Medical University, Xi'an 710032, China. [2]Department of General Surgery, 988 Hospital of Joint Logistic Support Force, Zheng Zhou 450000, China. [3]Specific Lab for Mitochondrial Plasticity Underlying Nervous System Diseases, National Demonstration Center for Experimental Preclinical Medicine Education, The Fourth Military Medical University, Xi'an 710032, China. [4]Department of pharmacy, 518 Hospital, Xi'an 710032, China. [5]These authors contributed equally: Yunhu Bai, Kenan Li. ✉e-mail: gdwkcy@163.com; wangyy@fmmu.edu.cn; yangyanl@fmmu.edu.cn

neurons has yet to be resolved. The manifestation of HE is the results of multiple pathogenic factors, including cerebral bioenergetics, mitochondrial dysfunction, oxidative stress and neuroinflammation[15–17]. Therefore, it is imperative to explore the pathophysiological mechanisms underlying HE.

This study initially compares the oxidative stress of SNr and cerebral cortex in two types of HE models of thioacetamide (TAA) and bile duct ligation (BDL) in mice, and then evaluates the effect of several mitochondria-centered anti-oxidative treatments on HE−induced bradykinesia. FOS-cre ERT2 (TRAP2) mice, together with applications of tamoxifen and viral vectors containing double floxed sites (DIO), are used to screen the brain area sensitive or susceptible to HE-induced hyperammonemia. DREADDs strategy is used to achieve the chemogenetic inhibition or activation of specific neuronal population under the statement of HE.

## Results

### TAA induced HE-like symptoms and systemic oxidative stress in mice

First, the effect of three doses of TAA (100, 150 and 200 mg/kg) on mice was screened. 150 mg/kg TAA could induce mice to be present with liver damage and clear bradykinesia, which is consistent with previous studies[18–20]. The higher dose of TAA could result in high mortality, while the lower dose of TAA could not induce bradykinesia in mice (Supplementary Fig. 1a–e). So 150 mg/kg TAA with consecutive 3 days was used for the following studies. It was found that TAA induced hepatic injury, which was reflected by the hepatocyte necrosis

by H&E staining (Fig. 1a), the increased levels of blood ALT, AST, and TBil (Fig. 1b–d), blood ammonia (Fig. 1e), as well as brain ammonia (Fig. 1f) by Elisa. Moreover, TAA also induced brain edema (Supplementary Fig. 2). Besides, TAA induced the systemic oxidative stress, which was reflected by the significantly decreased blood SOD and GPx levels (Fig. 1g, h), and influenced by the biomarkers of antioxidative activity. TAA mice displayed balance impairment by the rotarod test (Fig. 1i and see below), and the reduce of total distance and speed of spontaneous movement by the open field test (Fig. 1j, k). These data indicated that TAA induced HE-like symptoms and systemic oxidative stress in mice.

### TAA induced brain oxidative stress but SNr-specific mitochondrial dysfunction

SNr plays a pivotal role in mediating the motor deficit in HE[9,14]. Cerebral cortex is another target attacked by hyperammonemia and oxidative stress[21,22]. Considering this, detected the expression levels of autophagy markers (LC3B and PINK1), antioxidative stress markers (SOD1 and GPx1), mitochondrial uncoupling proteins (UCP2, UCP4, and UCP5), and mitochondrial factors including pro-mitochondrial fission factors of dynamin-related protein 1 (DRP1) and its phosphorylated version of p-DRP1, the mitochondrial fission factor (MFF) in charge of pro-midzone-type fission, the mitochondrial fission 1 (FIS1) in charge of pro-peripheral-type fission, and pro-mitochondrial fusion factor mitofusin 2 (MFN2), in SNr and cerebral cortex). As demonstrated by the western blot results, in SNr, TAA induced the significantly increasing levels of LC3B and PINK1, UCP2, as well as the significantly

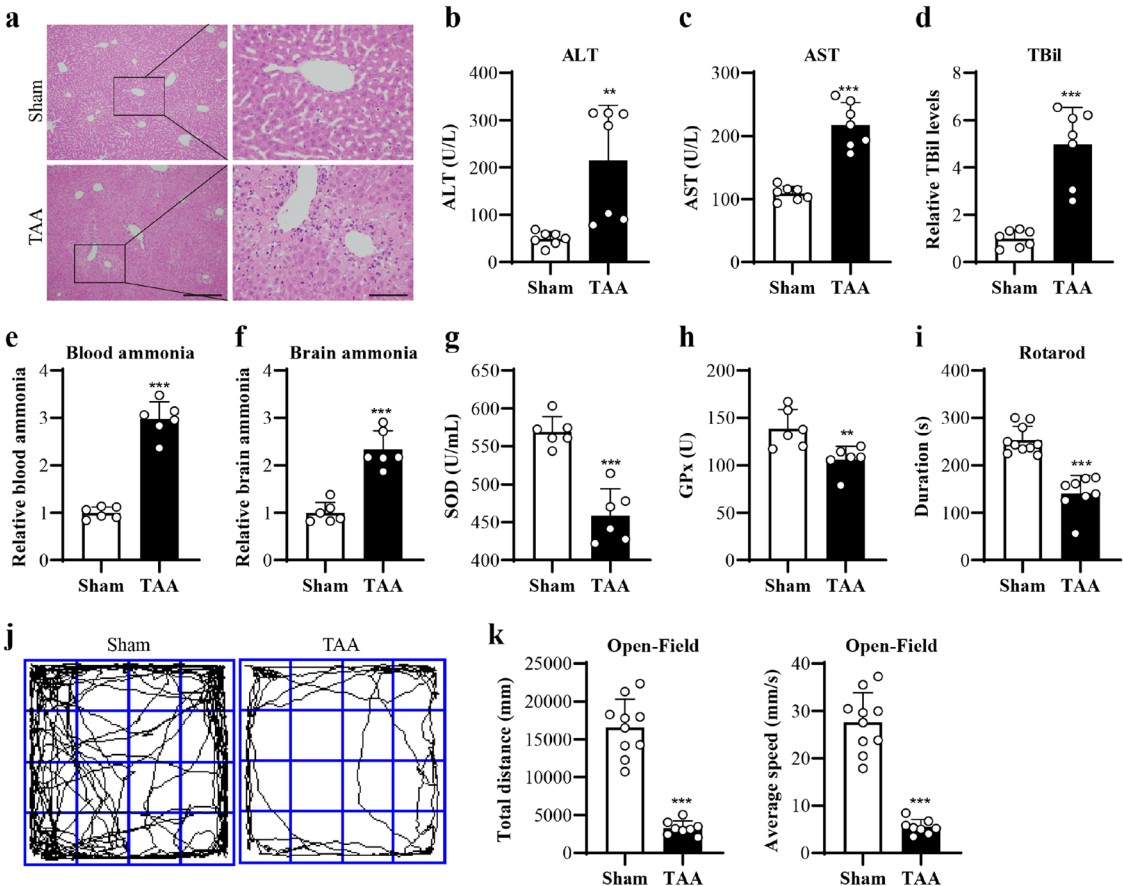

**Fig. 1 | TAA-treated mice exhibited HE-associated liver and behavioral changes. a** Liver H&E staining (left Bar = 100 μm, right Bar = 30 μm). **b−d** Elisa results of the serum levels of ALT, AST, and TBil (n = 7 per group). **e, f** Ammonia levels in blood and brain by Elisa (n = 6 per group). **g, h** Elisa results of antioxidative biomarkers of SOD and GPx in the blood (n = 6 per group). **i** Behavioral results of mice by rotarod test (n = 10, 8 per group). **j** The trajectory of spontaneous activity by open field test. **k** Analysis of total distance and speed of movement by open field test (n = 10, 8 per group). Data are presented as means ± SD. ** P < 0.01, *** P < 0.001 vs Sham. Two-tailed, unpaired, Student's t-test for all data. Source data are provided as a Source Data file.

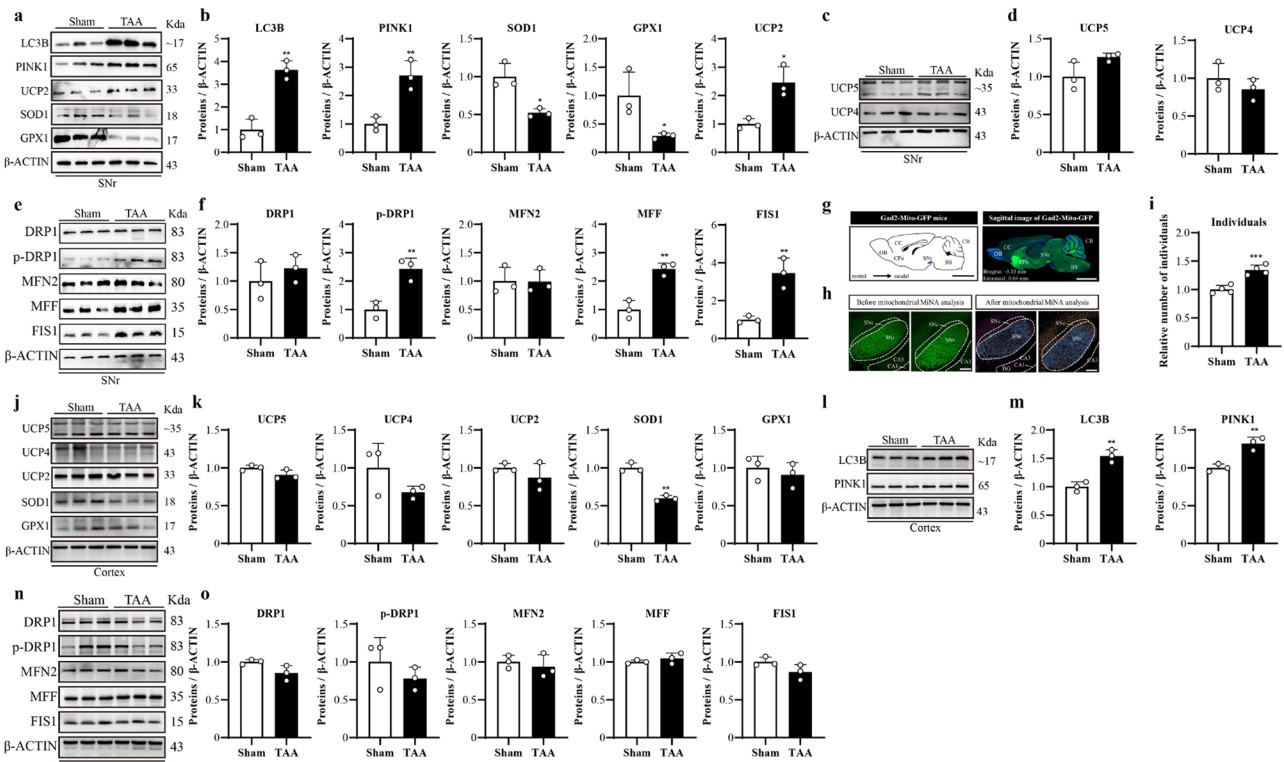

**Fig. 2 | TAA induced SNr neurons, but not cerebral cortex neurons, suffering an oxidative attack and mitochondrial dysfunction. a** SNr expression of LC3B, PINK1, UCP2, SOD1, and GPX1 from Sham and TAA, assayed by western blot. **b** Quantitative analysis of **a. c** SNr expression of UCP4 and UCP5 from Sham and TAA, assayed by western blot. **d** Quantitative analysis of **c. e** SNr expression of DRP1, p-DRP1(Ser616), MFN2, MFF, and FIS1 from Sham and TAA, assayed by western blot. **f** Quantitative analysis of **e. g** Schematic of a sagittal brain section and confocal images showing the highest bright GFP concentrated at OB, CC, CPu, SNr, CB, and BS from a GAD2-Mito-GFP transgenic mouse. (Bar = 1 mm). **h, i** Confocal images of mitochondrial GFP at SNr and its statistical results by MiNA analysis (n = 4 per

group, Bar = 100 μm). **j** Cortex expression of UCP2, UCP4, UCP5, SOD1, and GPX1 from Sham and TAA, assayed by western blot. **k** Quantitative analysis of **j. l** Cortex expression of LC3B and PINK1 from Sham and TAA, assayed by western blot. **m** Quantitative analysis of **l** (n = 3 per group). **n** Cortex expression of DRP1, p-DRP1(Ser616), MFN2, MFF, and FIS1 from Sham and TAA, assayed by western blot. **o** Quantitative analysis of **n.** Data are presented as means ± SD. *P < 0.05, **P < 0.01 vs Sham. n = 3 per group in western blotting. Two-tailed, unpaired, Student's t-test for all data. OB olfactory bulb, CC cerebral cortex, CPu caudate putamen (striatum), CB cerebellum, BS brainstem, SNr substantia nigra pars reticulate. Source data are provided as a Source Data file.

decreasing levels of SOD1 and GPX1 (Fig. 2a, b). TAA had no significant effect on UCP4 and UCP5 (Fig. 2c, d). Subsequently, mitochondrial morphology related proteins were observed, and the results showed that p-DRP1, MFF, and FIS1 expression increased in SNr^GAD2 neurons, while MFN2 had no significant changes (Fig. 2e, f). To further illustrated that mitochondrial fission occurred, the Mitochondrial Network Analysis tool (MiNA) analysis was performed directly to observe the morphological changes of mitochondrial network in SNr^GAD2 neurons. GAD2-Mito-GFP mice were used since all GAD2-expressing GABAergic populations in the central nervous system exhibited bright GFP fluorescence localized specifically to the mitochondrial compartment (Fig. 2g) in experimental mice. The MiNA analysis revealed that TAA induced a significant increase in the number of individual mitochondria in SNr (Fig. 2h, i). The western blot results showed that, in cerebral cortex, TAA induced the significantly decreasing levels of SOD1 and GPX1 (Fig. 2j, k), while increasing levels of LC3B and PINK1 (Fig. 2l, m). However, it had no effect on the expression of mitochondrial fission and fusion proteins (Fig. 2n, o). These data suggested that TAA selectively induced SNr neurons, leading to oxidative stress and mitochondrial dysfunction, rather than affecting cerebral cortex neurons.

## Treatment of mitoquinone (Mito-Q) ameliorated TAA-induced HE

Mitoquinone (Mito-Q) is a mitochondrial-targeting antioxidant drug, with several hundred-fold properties facilitating the

prevention of mitochondrial dysfunction, different from the untargeted antioxidants[23,24]. The study finding has reported that Mito-Q could dramatically alleviate the downregulation of Ucp2 and Ucp3 genes, ultimately improving the myocardial mitochondrial remodeling[23,25]. The effect of Mito-Q on TAA-induced HE has been detected, with the strategy shown in Fig. 3a. It was shown that Mito-Q treatment alleviated TAA-induced hepatocyte necrosis (Fig. 3b), and restored the levels of ALT (Fig. 3c), AST (Fig. 3d), blood ammonia (Fig. 3e), and brain ammonia (Fig. 3f) increased by TAA. Meanwhile, the Mito-Q treatment recovered the blood SOD (Fig. 3g) and GPx (Fig. 3h) levels decreased by TAA. The behavioral impairments including gait disorder (Fig. 3i), balance dysfunction (Fig. 3j), and the reduced mobility (Fig. 3k) in TAA-HE mice were alleviated significantly by Mito-Q treatment. Moreover, TAA−induced brain edema was alleviated considerably by Mito-Q treatment (Supplementary Fig. 2). The same dose of Mito-Q did not affect liver function (Supplementary Fig. 3a−c) or locomotor activities in normal mice (Supplementary Fig. 3d−f). Additionally, ROS levels within the SNr were detected by ROS probe in GAD2-Mito-GFP mice (Fig. 3l) to explore the effect of Mito-Q on SNr oxidative stress induced by TAA. The results showed the beneficial effect of Mito-Q on ROS level reduction within the SNr. It was obvious that HE mice receiving Mito-Q treatment were protected from TAA-induced oxidative stress (Fig. 3m), autophagy injury (Fig. 3n), and mitochondrial fragmentation impairment (Fig. 3o). Mito-Q treatment could ameliorate TAA-induced HE.

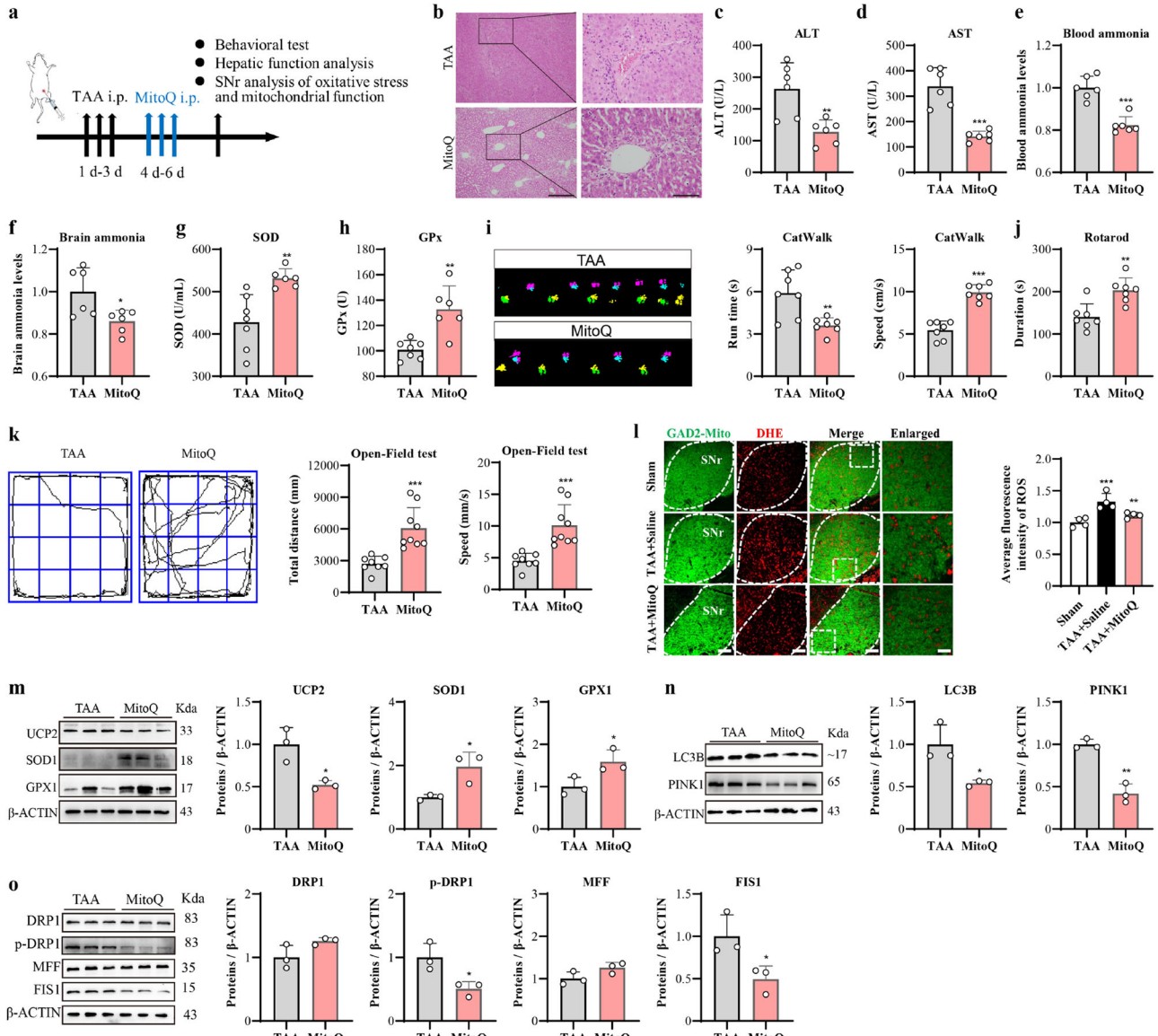

**Fig. 3 | Systemic treatment of Mito-Q ameliorated liver injury, bradykinesia, and mitochondrial dysfunction induced by TAA. a** Schematic flow chart to show the strategy. **b** H&E staining in TAA mice (TAA group) and Mito-Q-treated TAA mice (Mito-Q group) (Bar = 100 μm on the top and Bar = 30 μm on the bottom). **c, d** The serum levels of ALT and AST ($n = 6$ per group). **e, f** Ammonia levels in blood and brain ($n = 6$ per group). **g, h** Elisa results of antioxidative biomarkers of SOD and GPx in the blood ($n = 7, 6$ per group). **i** Schematic traces and analysis of the run time and average speed of the CatWalk test ($n = 7$). **j** Behavioral results of rotarod test ($n = 7$ per group). **k** Schematic traces and analysis of the total distance and average speed of open field test ($n = 8, 9$ per group). **l** ROS staining in the SNr of GAD2-Mito mice and quantitative analysis of ROS staining within SNr ($n = 4$ per group, Bar = 100 μm or 30 μm). **m** Western blot results and analysis of UCP2, SOD1, and GPX1 expression in SNr. **n** Western blot results and analysis of LC3B and PINK1 expression in SNr. **o** Western blot results and analysis of DRP1, p-DRP1(Ser616), MFN2, MFF, and FIS1 expression in SNr. Data are presented as means ± SD. *$P < 0.05$, **$P < 0.01$, ***$P < 0.001$ vs TAA. $n = 3$ per group in western blotting. Two-tailed, unpaired, Student's $t$-test for all data except one-way ANOVA with LSD's multiple comparison tests for l. Source data are provided as a Source Data file.

## Treatment of Mito-Q ameliorated BDL-induced HE

The effect of Mito-Q on BDL mice was studied. The strategy was presented in Fig. 4a. Consistent with previous reports[26,27], BDL induced mild hepatocytes necrosis (Fig. 4b), and increased the levels of ALT (Fig. 4c), AST (Fig. 4d), TBil (Fig. 4e), and blood ammonia (Fig. 4f). Specifically, BDL mice showed the balance impairment by rotarod test (Fig. 4g) and the bradykinesia was reflected as the decreasing total distance, average speed, and the number of activities in open field test (Fig. 4h, i). The similar deficiency of locomotor activity was observed from BDL mice group to 150 mg/kg TAA group. It was shown that Mito-Q treatment alleviated BDL-induced hepatic injury (Fig. 4b–f) and movement disorder (Fig. 4g–i), which was similar to the results from TAA mice. Western blot results showed that, in SNr, BDL induced the increase of UCP2, LC3B, PINK1, pDRP1, MFF and FIS1, but the decrease of SOD1 and MFN2 (Fig. 4j–o) . This confirmed the presence of oxidative stress and mitochondrial dynamic injury within the SNr in BDL mice. BDL mice receiving Mito-Q treatment were protected from oxidative stress, autophagy attack, and mitochondrial fragmentation (Fig. 4j–o), thus confirming the beneficial effect of Mito-Q on BDL model.

## TAA activated GAD2-expressing GABA population in medial SNr region

The targeted recombination in active populations (TRAP) approach was used to select access to neurons activated by specific stimuli so as to obtain genetic access to neurons activated by the HE-induced

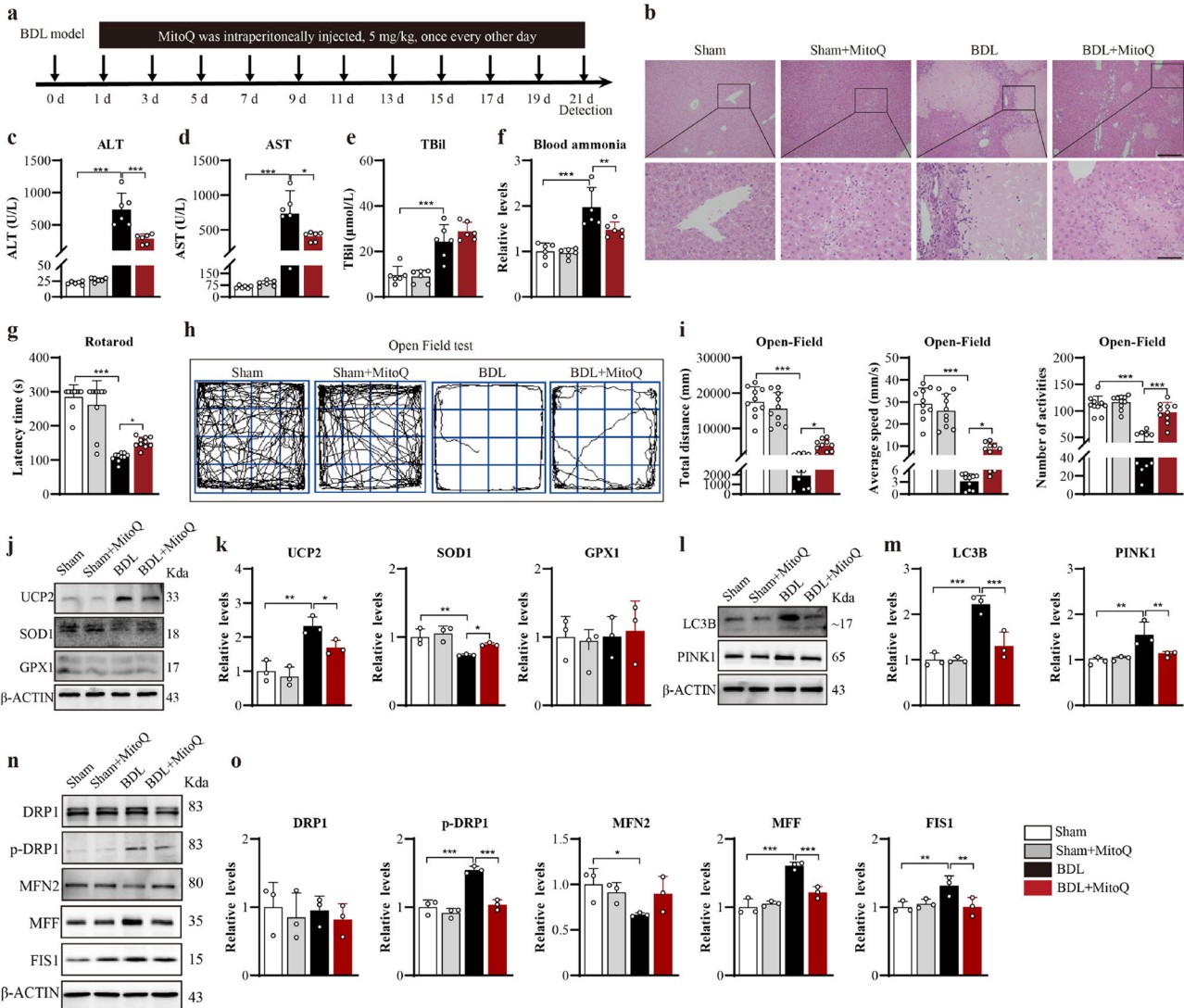

**Fig. 4 | Systemic treatment of Mito-Q ameliorated liver injury, bradykinesia, and mitochondrial dysfunction induced by BDL. a** Schematic flow chart to show the strategy. **b** H&E staining of BDL and Mito-Q (Bar = 100 μm or 30 μm). **c**–**e** The serum levels of ALT, AST, and TBil ($n = 6$ per group). **f** Ammonia levels in the blood ($n = 6$ per group). **g** Behavioral results of rotarod test ($n = 10$ per group). **h** Schematic traces of the open field test ($n = 10$ per group). **i** Analysis of the total distance, average speed, and the number of activities of the open field test ($n = 10$ per group). **j** SNr expression of UCP2, SOD1, and GPX1 from BDL and Mito-Q, assayed by western blot. **k** Quantitative analysis of **j**. **l** SNr expression of LC3B and PINK1 from BDL and Mito-Q, assayed by western blot. **m** Quantitative analysis of **l**. **n** SNr expression of DRP1, p-DRP1(Ser616), MFN2, MFF and FIS1 BDL and Mito-Q, assayed by western blot. **o** Quantitative analysis of **n**. Data are presented as means ± SD. *$P < 0.05$, **$P < 0.01$, ***$P < 0.001$ *vs* TAA. $n = 3$ per group in western blotting. One-way ANOVA with LSD's multiple comparison tests for all data. Source data are provided as a Source Data file.

hyperammonemia. FOS-cre ERT2 (TRAP2) mice receiving pAAV-hSyn-DIO-mCherry viral vectors on bilateral SNr were used to target the TRAPing to SNr region and limit the labeling to neuronal populations. Stereotactical injection was conducted on the first day (1 day) (Fig. 5a). After 14 days of recovery, the mice were injected with tamoxifen (TM) for 5 consecutive days, as an endeavor to stimulate active CreERT2-expressing cells and undergo Cre-mediated recombination (to be TRAPed), resulting in a permanent expression of the effector gene of mCherry. After 8-day intervals, the mice were treated by TAA to set up the HE models.

A dramatic increase of FOS+ neurons in total SNr was presented in the automated survey on coronal sections of sham-TRAPed and TAA-TRAPed distribution to SNr, as well as the statistical analysis (Fig. 5b, c). Furthermore, Nissl staining was used to count the number of neurons in SNr (Supplementary Fig. 4a, b). Liu DQ et al. reported two GABA populations with distinct spatial distributions, parvalbumin (PV)-expressing populations predominantly in the lateral SNr (lSNr), GAD2-expressing populations in the medial SNr (mSNr)[14]. GAD2 population, rather than PV population, was preferentially active in states of low motor activity. Then the FOS+ was investigated in both lSNr and mSNr. Surprisingly, mSNr saw a sharp increase of TAA-TRAPed cells, instead of lSNr (Fig. 5c). Furthermore, the mostly expressed type of markers of the FOS+ neurons were analyzed. The validity of the hSyn promoter was confirmed in Fig. 5d since most of the TAA-TRAPed cells were neurons rather than astrocytes. As is shown in Fig. 5e, 90% of the TRAPed cells were GABAergic (expressing GAD1, GAD2, or PV), the majority of which were expressed as GAD1 or GAD2 (81%), rather than PV (8%). Herein, TAA predominately activated GAD2-expressing GABA population at mSNr.

## Direct local high ammonia stimulation rapidly activated SNr GABA neurons

A key feature of hepatic encephalopathy is neurotoxic levels of ammonia in the brain[28]. The TRAP approach was used to map the

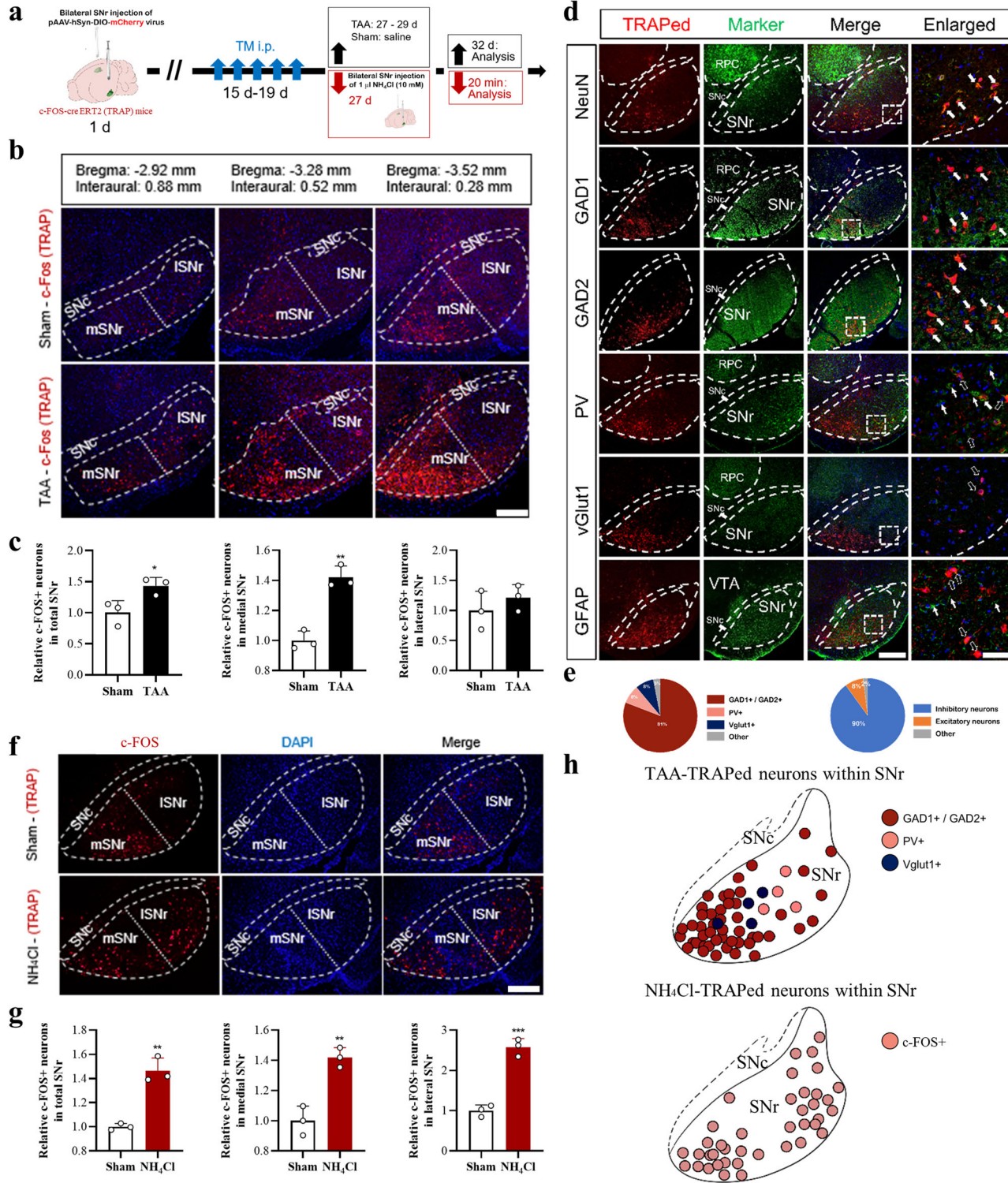

**Fig. 5 | TAA or direct local high ammonia stimulation activated SNr GABA neurons. a** Schematic flow chart to show the strategy. **b** Representative confocal images of TAA-TRAPed neurons in different layers of SNr (Bar = 100 μm). **c** Quantitative analysis of TAA-TRAPed neurons in total (left), medial (middle), and lateral SNr (right) (n = 3 per group, Bar = 100 μm). **d** Representative confocal images of TAA-TRAPed (red) neurons simultaneously expressing different makers (green) of NeuN, GAD1, GAD2, PV, vGlut1, and GFAP. Blue marked DAPI (Bar = 100 μm or 20 μm). **e** The charts represent the percentage of GAD1/ GAD2+, PV+, and vGluT1+ neurons, or the percentage of inhibitory and excitatory neurons. **f** Representative confocal images of NH4Cl-TRAPed neurons in SNr (Bar = 100 μm). **g** Quantitative analysis of NH4Cl-TRAPed neurons in total, medial, and lateral SNr

(n = 3 per group). **h** Schematic map of TAA-TRAPed (upper) and NH4Cl-TRAPed (lower) neurons within SNr. In the upper map, red, pink, and dark blue core rings represented GAD1+/GAD2+, PV+, and vGluT1+ neurons, respectively. Data are presented as means ± SD. \*P < 0.05, \*\*P < 0.01 vs with Sham. Two-tailed, unpaired, Student's t-test for all data. The arrows with dashed edges indicated the cells only expressing FOS+, thin solid arrows indicated the cells only expressing green fluorescence, and thick solid arrows indicated the cells expressing both and green fluorescences. SNc substantia nigra pars compacta, SNr substantia nigra pars reticulate, lSNr lateral SNr, mSNr medial SNr. Source data are provided as a Source Data file.

activated neurons under direct local high ammonia stimulation (NH₄Cl, 10 mM), with experimental procedures shown in Fig. 5a. The automated survey was performed after 20 min of ammonia application. The FOS+ mapping on the coronal sections in SNr was shown in Fig. 5f. According to the statistical analysis, direct local high ammonia stimulation rapidly activated SNr GABA neurons, which were distributed equally in both medial and lateral SNr (Fig. 5g). In the next phase, the TRAPed neurons activated by TAA and NH₄Cl within SNr were plotted (Fig. 5h). These results suggested the rapid activation of SNr GABA neurons by direct local high ammonia stimulation.

### Chemogenetic inhibition of SNr$^{GAD2}$ expressing GABA population ameliorated HE

The manipulation of specific cell populations was achieved by recently developed chemogenetic approaches. DREADDs are genetically modified muscarinic receptors without affinity for endogenous acetylcholine while the receptors can be activated by CNO (or DCZ), a pharmacologically inert metabolite of the atypical antipsychotic drug clozapine. DREADDs could couple via the Gq or Gi pathways to stimulate or inhibit neuronal activity, respectively. Two ligands (CNO and DCZ) were used to activate the hM4D (Gi) receptor and hM3D (Gq) receptor, with experimental procedures shown in Fig. 6a. Gi or Gq viral vectors were injected for chemogenetic manipulation. The confocal photographic picture of the viral vectors on the coronal section confirmed the correct stereotaxic localization (Fig. 6b, and Supplementary Fig. 5a). The patch clamp recordings were used to demonstrate the electrophysiological features of red mCherry-expressing SNr neurons in the Gi group. The results confirmed the decreasing firing of these chemogenetically inhibited neurons (Fig. 6c). There was a sharp decrease of the firing rate of GAD2-expressing GABA population in CNO-post group when compared with that in CNO-pre group (Fig. 6d). Open field results confirmed the locomotor improvement in Gi group (Fig. 6e, f, and Supplementary Fig. 5b, c). Figure 6g and Supplementary Fig. 5d showed the improvement of behavioral balance in the TAA group treated by Gi.

### Chemogenetic activation of SNr$^{Gad2}$ expressing GABA population deteriorated HE

In the next phase, the effect of opposite chemogenetical (Gq) manipulation on TAA mice was studied. DREADD viral vector of AAV8-hSyn-DIO-hM3D(Gq)-mcherry was used for chemogenetic activation. The patch clamp recordings confirmed the increased firing of the neurons chemogenetically activated by Gq in CNO-post group (Fig. 6h, i). Moreover, the open field results confirmed the locomotor deterioration in TAA + Gq group by CNO and DCZ (Fig. 6j, k, and Supplementary Fig. 5b, c). The rotarod test confirmed the deterioration of behavioral balance in the TAA + Gq group by CNO and DCZ (Fig. 6l, and Supplementary Fig. 5d, e).

In addition, the expression levels of the antioxidant kinases, including UCP2, SOD1 and GPX1, and autophagy markers, including LC3B and PINK1, were detected by western blot. It was shown that Gi inhibition of SNr$^{GAD2}$ expressing GABA population could not only ameliorate HE, but also increase UCP2 expression level (Fig. 6m). Moreover, Gq activation could not only deteriorate HE, but also decrease UCP2 expression level (Fig. 6m). There were no significant changes in SOD1 and GPX1, as well as LC3B and PINK1 in either Gi group or Gq group (Supplementary Fig. 6a, b).

### Targeted UCP2 OE on SNr$^{GAD2}$ population ameliorated TAA-induced HE

Further, the protective effect of UCP2 on HE behavior in SNr$^{GAD2}$ neurons by regulating oxidative stress and mitochondrial dynamics was studied. As a mitochondrial protein, UCP2 plays a critical role in mitochondrial function, including separating oxidative phosphorylation from ATP synthesis and controlling proton reentry into the mitochondrial matrix[29]. Sustained UCP2-mediated uncoupling reduces mitochondrial ROS production to protect neurons[30]. UCP2 is involved in the regulation of the activity of neurons and astrocytes (glial cells) in several brain areas[30,31]. Next, an on / off-switching cassette for UCP2 was used to express UCP2 by recombinase Cre-mediated excisional deletion of a spacer DNA flanked by DIO on GAD2-iris-cre mice. UCP2 viral vectors were injected into the bilateral SNr of GAD2-ires-cre mice to selectively target GAD2-expressing GABA population (Fig. 7a). Western blot results showed there was a significant increase (56.36%) or decrease (50.29%) of UCP2 levels after *Ucp2* overexpression (OE) or knock-down (KD) treatment, respectively (Fig. 7b). *Ucp2* OE treatment alleviated bradykinesia in TAA mice, while *Ucp2* KD treatment aggravated bradykinesia (Fig. 7c, d). Moreover, the same dose of UCP2 AAV virus did not affect liver function (Supplementary Fig. 3a–c) or behavioral movement (Supplementary Fig. 3d–f) in normal mice. Furthermore, it was shown that *Ucp2* OE treatment alleviated oxidative stress (Fig. 7e, f), autophagy (Fig. 7g, h), and mitochondrial fragmentation (Fig. 7i, j). On the contrary, *Ucp2* KD treatment aggravated oxidative stress, autophagy, and mitochondrial fragmentation. These results suggested the targeted *Ucp2* OE on SNr$^{GAD2}$ population ameliorated HE.

### Both Mito-Q and targeted UCP2 upregulation rescued basal metabolism disorder and mitochondrial respiration dysfunction in HE mice

A recent study confirmed that metabolic disturbances, oxidative stress, and defects in mitochondrial respiratory function may affect liver injury and neuron injury[32]. It was hypothesized that HE caused changes in the metabolic phenotype of mice, and antioxidant stress improved metabolism by protecting mitochondria. Metabolic substrates were detected through respiratory metabolism in the metabolic cage were detected to characterize energy metabolism in HE mice. The total respiratory exchange ratio (RER) increased considerably and recovered significantly after Mito-Q administration, although there was no discernible difference in oxygen consumption (VO₂) and carbon dioxide production (VCO₂) (Fig. 8a–c, and Supplementary Fig. 7a–c). Moreover, the result of Y total activity (YTOT) showed that TAA caused a significant decrease in activity, whereas Mito-Q had no significant change (Fig. 8d, and Supplementary Fig. 7d). It should be noted that HE mice experienced a lower body temperature which was restored after Mito-Q treatment (Fig. 8e).

Finally, the mitochondrial respiratory State III, State IV, and RCR were assessed using the isolated mitochondria from SNr. The mitochondrial respiratory chain complex III, IV, and ATP were assessed using total protein isolated from SNr. Mito-Q and *Ucp2*OE recovered the impairments of mitochondrial respiratory state III, RCR, and complex III/IV induced by TAA, but no significant difference in respiration of State IV (Fig. 8f–j). The mitochondrial respiratory chain complex III, IV, and ATP were assessed using total protein isolated from SNr. Mito-Q and Ucp2 OE recovered the impairments of complex III / IV induced by TAA (Fig. 8k, l). Mito-Q restored the decreased ATP synthesis capacity of mitochondria induced by TAA (Fig. 8m). These data indicated the beneficial effect of Mito-Q and UCP2 upregulation on metabolism disorders and mitochondrial dysfunction in HE mice.

## Discussion

Based on the present results, we proposed a model in which mitochondria could be the center for pathogenesis of hepatic encephalopathy (HE) (Fig. 9). The steps involved in HE include: firstly, liver injury causes hyperammonemia; secondly, hyperammonemia causes oxidative stress in both liver and brain; thirdly, the mitochondria located in the GAD2-expressing population in SNr are susceptible to oxidative stress attack, making SNr becomes the target of hyperammonemia; fourthly, mitochondrial dysfunction in SNr$^{GAD2}$ population results in the hyperactivation of neuronal cells; fifthly, increased of GABAergic

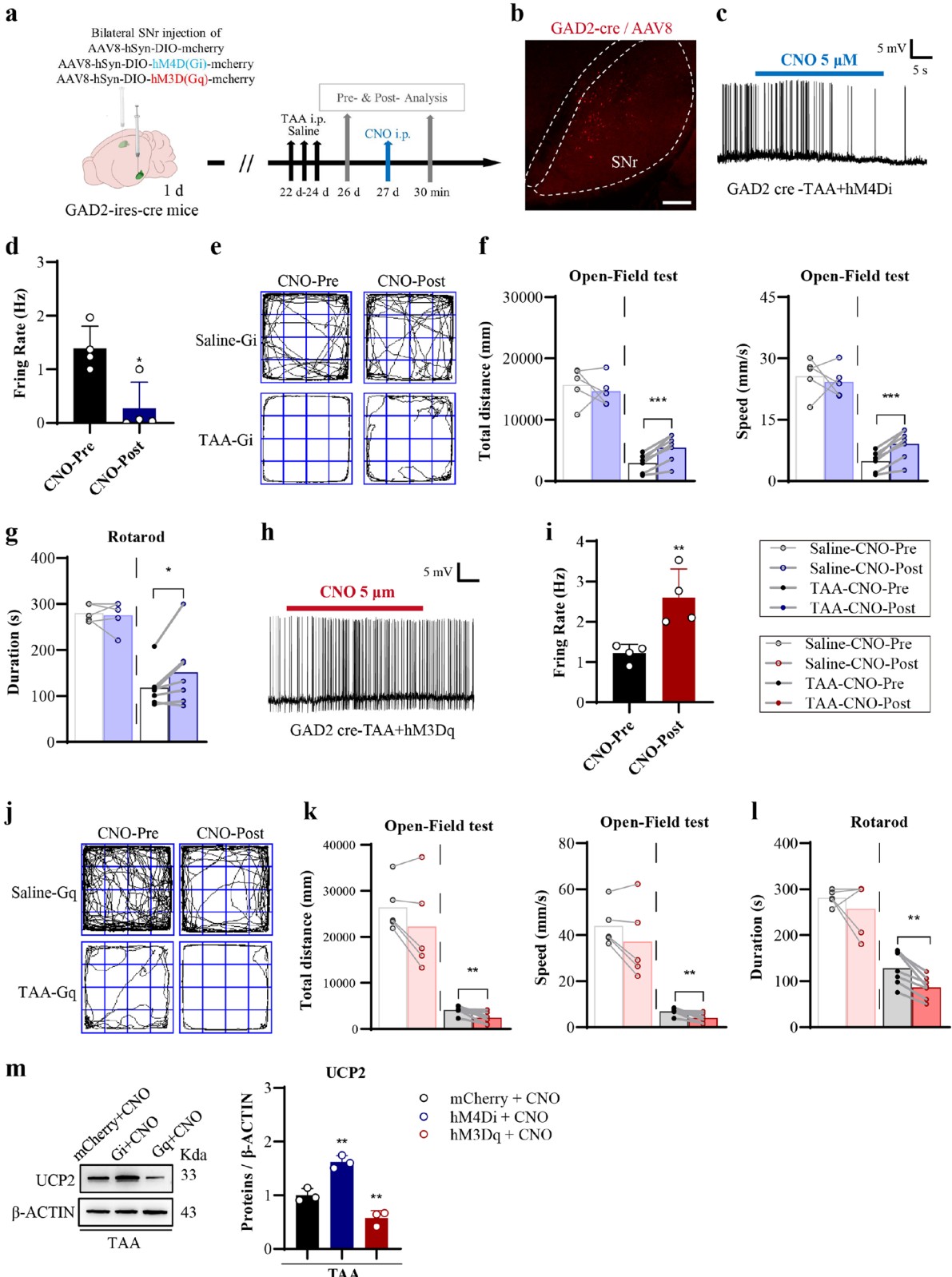

tone leads to hepatic encephalopathy. Additionally, the systemic administration of mitochondrial-targeted antioxidant Mito-Q may reduce liver injury induced by both TAA and BDL, thus improving HE (green evidence chain on the left). Significantly, chemogenetic inhibition of SNr$^{GAD2}$ expressing GABA population (Gi) or targeted over-expression of mitochondrial uncoupling protein 2 (Ucp2 OE) in such population could ameliorate neuronal oxidative stress and

mitochondrial dysfunction, and finally reverse HE-induced bradyki-nesia (green evidence chain on the right).

Hyperammonemia has been linked to oxidative stress in the central nervous system of patients with HE, as well as mitochondrial dysfunction[33,34]. Oxidative stress aggravates the cerebral effects of ammonia in patients with liver disease[16,35]. Moreover, oxidative stress has been associated with ammonia toxicity, characterized by excessive

**Fig. 6 | Causal effects of SNr^GAD2 neuron activity on motor behavior in HE.**
**a** Schematic flow chart of Gi or activation Gq of GAD2-expressing GABA population in SNr of GAD2-ires-cre mice. **b** Representative confocal images of confirmation of AAV8 virus located at SNr (Bar = 100 μm). Regions of interest on 4 sections from a series of section per mouse. Three mice were used. **c**, **d** Electrophysiological recording of spontaneous firing frequency (**c**) and statistical analysis (**d**) of firing rate (Hz) of GAD2-expressing GABA population in SNr with hM4Di chemogenetic inhibition (n = 4 per group, Bar = 5 mV /5 s). **e** The trajectories by open field test of the saline-Gi or TAA-Gi mice with pre- and post-CNO injection. **f** Behavioral results of the total distance and speed by open field test (n = 5, 7 per group). **g** Behavioral results of duration on the rotarod test (n = 5, 7 per group). **h**, **i** Electrophysiological recording of spontaneous firing frequency (**h**) and analysis (**i**) of firing rate (Hz) of GAD2-expressing GABA population in SNr with hM3Dq chemogenetic activation (n = 4, Bar = 5 mV/5 s). **j** The trajectories by open field test of the saline-Gq or TAA-Gq mice with Pre- and Post-CNO injection. **k** Behavioral results of the total distance and speed by open field test (n = 5, 7 per group). **l** Behavioral results of duration on the rotarod test (n = 5, 7 per group). **m** Western blot results and analysis of UCP2 level in SNr of control (mCherry + CNO), chemogenetic inhibition (Gi + CNO), and chemogenetic activation (Gq + CNO) groups (n = 3). Data are presented as means ± SD. *P < 0.05, **P < 0.01, ***P < 0.001 vs with CNO-Pre or mCherry + CNO. Two-tailed, unpaired, Student's t-test for **d**, **i**. Two-tailed, paired, Student's t-test for **f**, **j**, **k**, **l**. One-way ANOVA with LSD's multiple comparison tests for m. Gq hM3Dq, Gi hM4Di. Source data are provided as a Source Data file.

ROS / RNS production via endogenous mechanisms or impaired detoxification function[16,36]. Ammonia is well known for inducing direct oxidative stress in neurons, resulting in ROS production and abnormal activity[37]. Nonetheless, Jayakumar et al. reported that the direct application of glutamine resulted in the generation of free radicals only in astrocytes, but not in neurons[38]. The majority of the neuron-related researches are in vitro, while directly targeting the regulation of mitochondrial oxidative stress of specific neurons was observed in vivo. In the present study, inhibition of SNr GABA neurons alleviated HE-induced hypokinesia by increasing mitochondrial UCP2 expression level, stabilizing mitochondrial fission and respiratory chain complex, and enhancing oxidative defense. According to the findings, neurons are vulnerable to oxidative stress injury.

Montes-Cortes et al. have compared the level of the oxidative marker malondialdehyde (MDA) in hepatic encephalopathy patients with chronic liver disease and in healthy volunteers[39]. The results have confirmed that the hyperammonemia in HE has been accompanied with the increasing MDA. Sfarti C et al. have performed a prospective case-control study in which 40 patients have been divided into two groups: group A consisted of 20 cirrhotic patients with both HE and increased systemic ammoniemia; group B consisted of 20 cirrhotic patients with normal systemic ammoniemia and without HE[40]. The activity of SOD, GPx, MDA, and ammoniemia were evaluated. HE patients in group A showed significant decreases in SOD and GPx activities, and significant increases of MDA levels. Moreover, there is a significant correlation between the main oxidative stress markers and the levels of systemic ammonia. Görg et al. observed the significantly increasing levels of other two oxidative markers of heme oxygenase 1 (HO1) and NADPH oxidase 4 (NOX4) in post-mortem brain samples from HE patients[16]. All these clinical data have emphasized the influence of oxidative stress in HE pathogenesis in cirrhotic patients.

In the present study, Mito-Q, a mitochondria-targeted antioxidant drug, was used. Mito-Q can be distributed in the mitochondrial matrix facing the surface of the inner mitochondrial membrane, where it is optimally positioned to reduce mitochondrial-resourced ROS[41]. Mito-Q is important for the central nervous system since it could rapidly cross through the blood-brain barrier and accumulate in the brain[42,43]. In fact, Mito-Q has been used safely used in Phase II clinical trials for patients with Parkinson's disease patients and severe liver injury patients[44,45]. In the present research, Mito-Q could improve liver damage, metabolism impairment, and mitochondrial dysfunction induced by TAA or BDL. Previous studies have reported the reduction of cerebral edema in HE patients by the systematic use of Mito-Q, which may be attributed to the functional recovery or remodeling of the communication between astrocytes and neurons. The present results show the beneficial effect of the targeted overexpression of UCP2 on the rescue of liver damage, metabolism impairment, and mitochondrial dysfunction in TAA mice. These results have indicated the key role of the communication of the central nervous system and peripheral tissues in different types of encephalopathy resulting from the injuries of liver, gut, kidney, and so on. The remission of the injuries

of peripheral tissues could promote the recovery of neuronal function in the brain. Meanwhile, neuronal homeostasis may also contribute to the functional maintenance of peripheral tissues. Given that, more attention should be paid to the reciprocal effect of neurons and hepatic cells in the clinical treatment of HE.

The present study has confirmed the hypothesis of GABAergic tone in HE. A growing number of experimental studies (in vivo or in vitro) have reported the functional changes of SNr GABAergic neurons in different types of encephalopathy or neurological diseases, especially in Parkinson's disease and epilepsy[46,47]. Moreover, it has been shown that targeting inhibition the GAD2 neuron of SNr can significantly improve motor function after injury[14]. The present results have shown that both HE and direct high ammonia can activate SNr GABA population, especially the GAD2-expressing GABA population in the medial SNr region. Chemogenetic inhibition of such a population could improve bradykinesia performance in HE mice, possibly by relieving oxidative stress and mitochondrial dysfunction in GAD2 neurons. Conversely, chemogenetic activation of such population could worsen bradykinesia, possibly by aggravating oxidative stress and mitochondrial dysfunction. These results emphasize the role of GABAergic tone in HE.

The current study also reexamined the significance of bradykinesia-like behavioral change as a clinical indication. Bradykinesia, or hypokinesia, or decreased behavioral or locomotor activity, is a major motoric alteration in HE patients, which is attributed to the changes in neuronal circuits between the basal ganglia and the prefrontal cortex[48,49]. At the normal statement, neuronal circuits involved in motor activity include the nucleus accumben → ventral pallidum → mediodorsal thalamus → prefrontal cortex[50]. Previous reports have indicated that in HE situation, motor activity is modulated via different neuronal circuits than in normal statements. In the HE situation, the neuronal circuits involved in hypokinesia include the nucleus accumben → SNr → ventromedial thalamus and prefrontal cortex[50]. Therefore, SNr is critical for locomotor dysfunction in HE. The current research indicates the hyperammonemia level could directly activate GABAergic neurons within the SNr, reflecting as the expression of activation marker FOS. These activated neurons may be representative of the hyperactivation state. It has been revealed in the present study that the chemogenetic inhibition of such population exerts the beneficial effect on the oxidative stress, mitochondrial dysfunction and bradykinesia disorder in HE mice. on the contrary, the chemogenetic activation of such population exerts the worse effect on the oxidative stress, mitochondrial dysfunction and bradykinesia disorder in HE mice. These results suggest imply the validity of increased GABA tone theory in pathological mechanisms underlying HE.

There are three limitations are as follows: firstly, there are some experimental procedures which are only done in TAA HE model, but not in BDL HE model; secondly, the involvement of excitatory neuronal population and glial cells in HE pathology is not studied; thirdly, the detailed changes of the downstream of the activated SNr^GAD2 neurons in HE are not clarified.

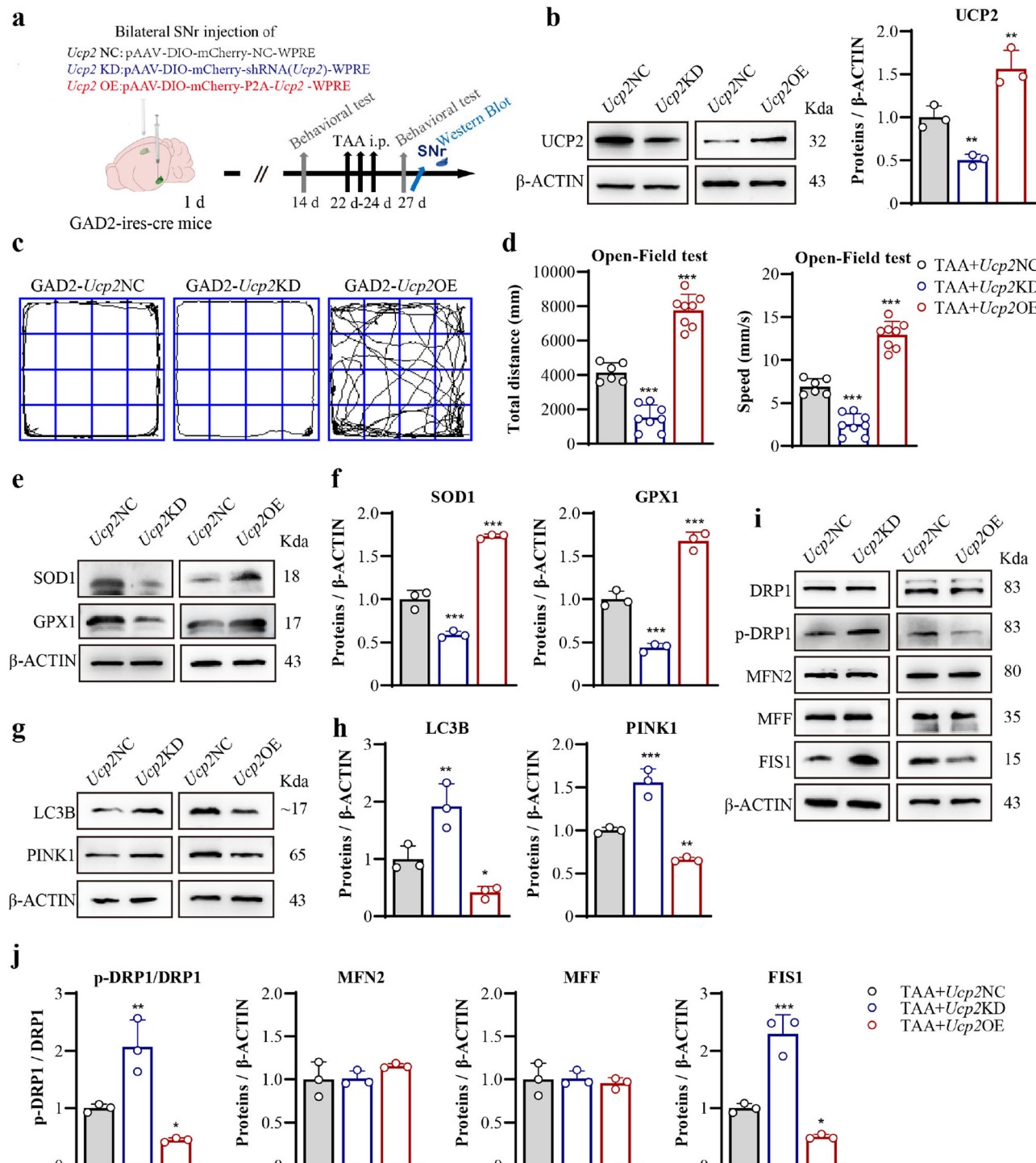

**Fig. 7 | Effects of UCP2 on the behavior of GAD2 mice and mitochondria of GAD2 neurons in SNr of HE. a** Schematic flow chart of *Ucp2* NC, *Ucp2* KD and *Ucp2* OE of SNr^GAD2 expressing GABA population on GAD2-ires-cre mice. **b** SNr expression of UCP2 from *Ucp2* NC, *Ucp2* KD and *Ucp2* OE assayed by western blot. **c** The trajectories of *Ucp2* NC, *Ucp2* KD and *Ucp2* OE GAD2-ires-cre mice by open field test (*n* = 6, 8, 8 per group). **d** Results of the total distance and speed by open field test (*n* = 6, 8, 8 per group). **e** SNr expression of a mitochondrial antioxidative marker of SOD1 and GPX1 from *Ucp2* NC, *Ucp2* KD and *Ucp2* OE, assayed by western blot.

**f** Quantitative analysis of **e. g** SNr expression of LC3B and PINK1 from *Ucp2* NC, *Ucp2* KD and *Ucp2* OE, assayed by western blot. **h** Quantitative analysis of **g. i** SNr expression of DRP1, p-DRP1(Ser616), MFN2, MFF, and FIS1 from *Ucp2* NC, *Ucp2* KD and *Ucp2* OE, assayed by western blot. **j** Quantitative analysis of **i.** Data are presented as means ± SD. *$P < 0.05$, **$P < 0.01$, ***$P < 0.001$ *vs* with TAA + *Ucp2* NC. *Ucp2* NC samples were collected from the same batch of samples in western blotting (*n* = 3 per group in all). One-way ANOVA with LSD's multiple comparison tests for all data. Source data are provided as a Source Data file.

The current investigation shows that HE or local high ammonia stimulation could directly activate the GAD2-expressing GABA population in the medial SNr region. Additionally, the chemogenetic inhibition of this population, or the targeted overexpression of mitochondrial UCP2 in such population, or systemic application of a mitochondrial-targeting antioxidant drug Mito-Q could effectively ameliorate HE. It is indicated that the mitochondria in the SNr^GAD2 population may be sensitive to ammonia imbalance and could be seen

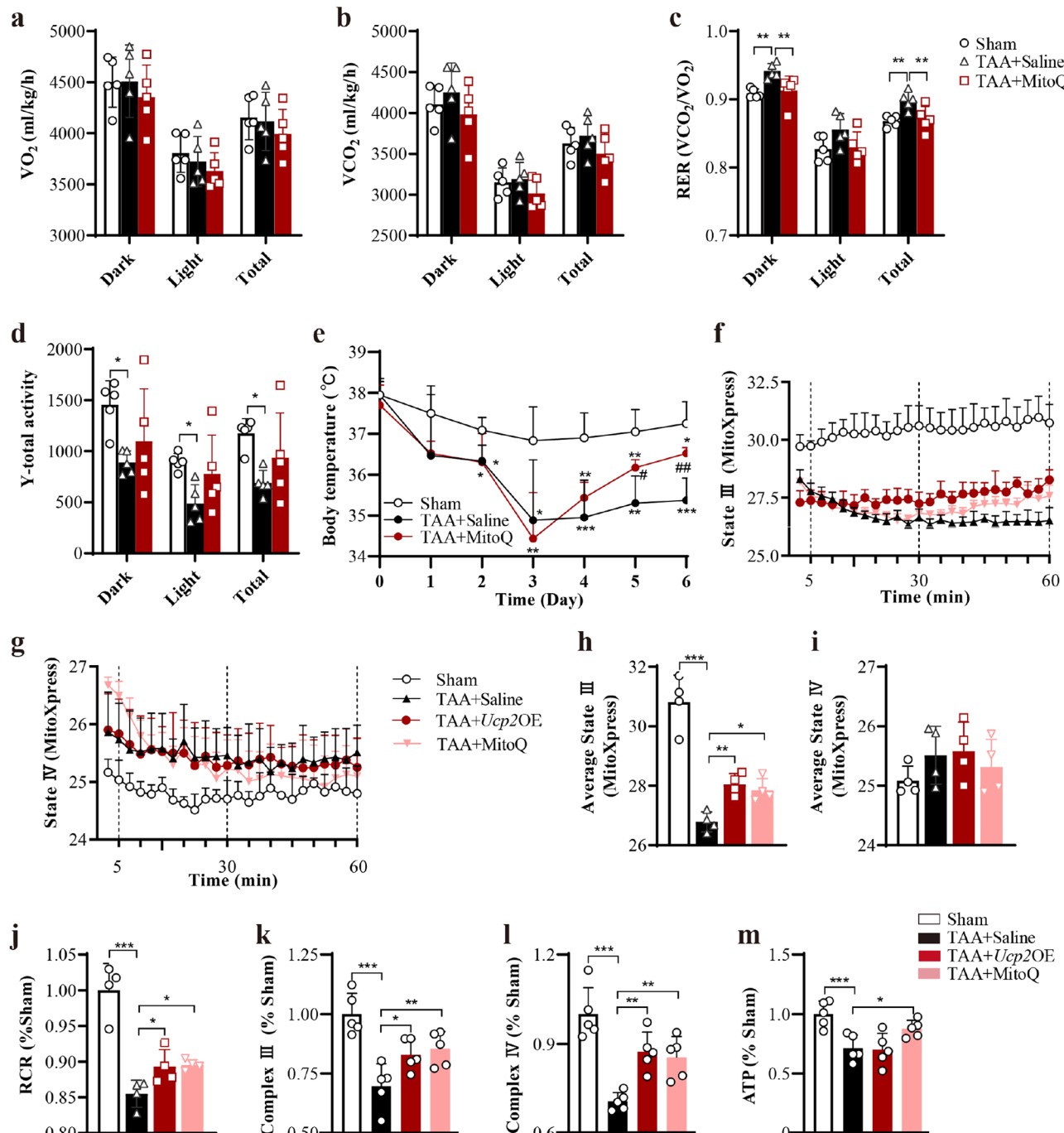

**Fig. 8 | Mito-Q and *Ucp2*OE improved mitochondrial respiration in SNr of HE mice. a–d** Analysis of VO$_2$ (**a**), VCO$_2$ (**b**), RER (**c**), and Y-total activity (**d**) during the dark, during the light, and total test period in Sham, TAA + Saline, and TAA + Mito-Q groups (*n* = 5 per group). **e** Mito-Q alleviated TAA-induced hypothermia (*n* = 6 per group). **f, g** The temporal changes of mitochondrial State III (**f**) and State IV (**g**) respiration in SNr in different groups (*n* = 4 per group). **h, i** MitoXpress of average State III (**h**) and State IV (**i**) respiration in SNr in different groups (*n* = 4 per group). **j** Mitochondrial respiratory control rate (RCR) in SNr in different groups (*n* = 4 per

group). **k, l** Activity of mitochondrial respiratory chain complex III (**k**) and IV (**l**) in SNr in different groups (*n* = 5 per group). **m** ATP levels in SNr in different groups. Data were presented as means ± SD. *$P < 0.05$, **$P < 0.01$, ***$P < 0.001$ *vs* with Sham or TAA + Saline. One-way ANOVA with LSD's multiple comparison tests for all data. VO$_2$ oxygen consumption, VCO$_2$ carbon dioxide production, RER respiratory exchange ratio, RCR respiratory control rate. Source data are provided as a Source Data file.

as the center for the treatment of HE. Present study provides a perspective into the relationship between mitochondria and HE.

## Methods
### Animal
The present study used adult (6–8 w) male wild-type C57BL/6 J, FOS-cre ERT2 (TRAP2), GAD2-ires-cre, GFP-Mito tag$^{floxed}$, and GAD2-Mito-

GFP mice. Specifically, C57BL/6 J mice were purchased from the Fourth Military Medical University Animal Center (Xi'an, China). All mice were housed in a 12 h light (8 am)/dark (8 pm) environment. Male mice were used throughout the study to further control the differences caused by animal sex differences. All experiments in this study were performed following the ethical guidelines of the International Association for the Study of Pain and approved by the Pain Research Committee of the

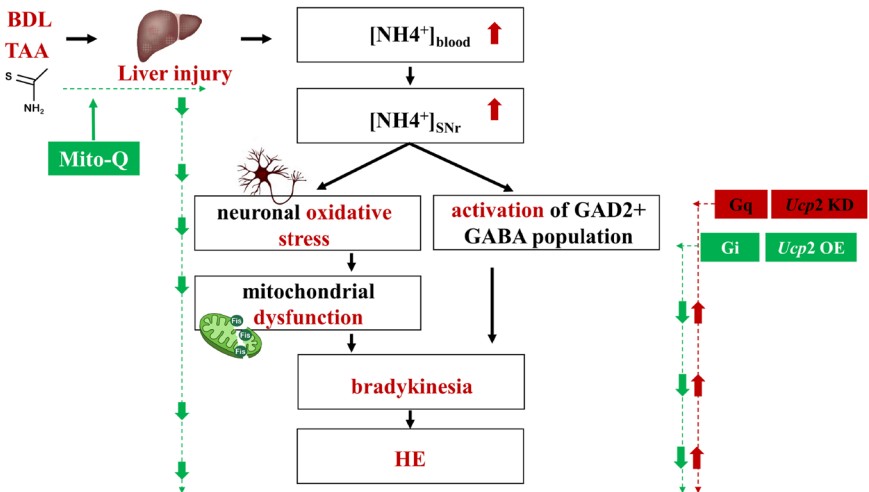

**Fig. 9 | Model illustrating SNr GAD2-expressing population neurons mechanism regulated by mitochondrial in HE bradykinesia.** The green arrow path represented the protection role. The red arrow path represented the deterioration role. BDL bile duct ligation, HE hepatic encephalopathy, Gq hM3Dq, Gi hM4Di, SNr substantia nigra pars reticulate. Source data are provided as a Source Data file.

Fourth Military Medical University. All mice were randomly assigned to the groups.

FOS-cre ERT2 (TRAP2) mice were purchased from Jackson Laboratory (FOS[tm2.1(icre/ERT2) Luo/J]; stock #030323), which was designed to express tamoxifen (TM)−inducible, enhanced Cre recombinase (icre/ERT2) from the FOS promoter/enhancer elements, without interfering with the expression of endogenous Fos. TRAP2 mice were an effective Cre-lox tool facilitating inducible iCre recombination in Fos-expressing cells/tissues. FOS positive expression (FOS+) represented the neurons activated by specific stimuli. TRAP2 mice, integrated with the infection by viral vectors containing double floxed sites (DIO) (AAV8-hSyn-DIO-mCherry) indicated by mCherry fluorescence may result in FOS+ neurons that express mCherry. Genotype identification was performed with primers GTC CGG TTC CTT CTA TGC AG; GAA CCT TCG AGG GAA GAC G and CCT TGC AAA AGT ATT ACA TCA CG.

GAD2-ires-cre mice were purchased from Jackson Laboratory and exhibited internal ribosome entry site and cre recombinase in the 3' UTR of the glutamic acid decarboxylase 2 locus (GAD2). As such, the endogenous GAD2 promoter/enhancer elements target the GABAergic population in the central nervous system represented by GAD2 positive neurons for cre expression. When the GAD2-ires-cre mice were infected with a viral vector containing double-floxed inverse open reading frames (DIO), including pAAV-DIO-mCherry-NC-WPRE (*Ucp2* NC); pAAV-DIO-mCherry-shRNA (*Ucp2*)-WPRE (*Ucp2* KD); pAAV-DIO-mCherry-P2A-*Ucp2*-WPRE (*Ucp2* OE), the GAD2-expressing GABA population could achieve the overexpression (OE) or knock-down (KD) regulation. The primers CTTCTTCCGCATGGTCATCT, CACCC-CACTGGTTTTGATTT, and AAAGCAATAGCATCACAAATTTCA were employed for genotype identification.

Additionally, when the GAD2-ires-cre mice were infected with a viral vector containing the elements of designer receptors exclusively activated by designer drugs (DREADDs) including hM4D or hM3Dq (AAV8-hSyn-DIO-hM4Di (Gi)-mCherry, AAV8-hSyn-DIO-hM3Dq (Gq)-mCherry), the GAD2-expressing GABA population in the special area could achieve chemogenetic inhibition (Gi) or activation (Gq), respectively.

GFP-Mito tag[floxed] mice were generated by GemPharmatech Co., Ltd (Nanjing, China), in which a Mito-Tag cassette of CAG-LSL-GFP-Mito tag was floxed in accordance with the study[51]. A construct containing CAG-loxP-STOP-loxP-Kozak-GFP-Mito-TGA-pA was targeted to Rosa26 of the mouse Gt (ROSA) 26 S. When GFP-Mito tagfloxed mouse crossed with GAD2-iris-cre drivers, the resulting mouse line, referred

to as GAD2-Mito-GFP, exhibited bright mito-GFP fluorescence localized specifically to the mitochondrial compartment of all GAD2-expressing GABAergic population in the central nervous. Primer Rosa26 - tF1: CCCAAAGTCGCTCTGAGTTGTTA. The CAG−tR1: TGGCGTTATCTATGGGAACATACGTC; Rosa26-tR1: TCGGGTGAGCATGTCTTTAATCT was used for genotype identification.

### Establishment of hepatic encephalopathy (HE) model
Both TAA model and BDL model were adopted as hepatic encephalopathy (HE) models in the present study.

TAA model: A toxin thioacetamide (TAA) (Sigma, Lot#BCCD3910, USA) model is used to induce HE[19,20]. Briefly, intraperitoneal (i.p.) injections of TAA (150 mg per kg of body weight) were administered once per day for consecutive 3 days. The Sham control group was administered with the same volume of saline. Subcutaneous injection of 25 mL/kg 77.0 mM NaCl, 227.5 mM dextrose, and 20.0 mM KCl was administered 12 h after TAA injection for the first time to prevent hypoglycemia and electrolyte imbalance. After successful modeling, intraperitoneal injections of Mitoquinone mesylate (Mito-Q, MCE, HY-100116A) 5 mg/kg were administered for three consecutive days. Similarly, the control group was given the same dose of solvent.

Bile duct ligation (BDL) model: Sodium pentobarbital (40 mg/kg, i.p) was used as anesthesia for the male mice (10−12 w) during the BDL surgery. After skin preparation, the skin was sterilized by using 70% ethanol, and an incision of about 1.5 cm was made along the ventral median line. The bile duct was separated from the portal vein and hepatic artery and dissociated for a certain length after opening the abdomen. Moreover, the 5−0 suture was knotted at the upper end of the common bile duct. The bile duct at the upper margin of the pancreas was ligated again to ensure that it was effectively blocked without being cut off. The abdominal organs were rinsed by using 0.9% NaCl solution and put back in situ. The abdominal cavity was then sutured and disinfected. This section of the experiment was divided into four groups: Sham operation was performed in the sham group, that is, free bile duct alone; Sham + Mito-Q group was given an intraperitoneal injection of Mito-Q (5 mg/kg, once every other day) after sham operation; BDL operation group and BDL + Mito-Q group, all mice were observed until the third week.

### SNr isolation
The mice were euthanized by deeply anesthetized with pentobarbital sodium (100 mg/kg, i.p.) and then cervical dislocation. Brain tissue was

removed from −80 °C to −20 °C for about 20–30 min (Supplementary Fig. 8a). The metal bracket, brain mold, and sampling instrument were placed on ice to cool (Supplementary Fig. 8b) before placed on the fine coronal meninges of mice (at this time, the brain tissue was hard and convenient for cutting, Supplementary Fig. 8c). Then, the brain was quickly placed ventrally downward for SNr. Meanwhile, the razor blade was used to make the first cut at the cerebellar junction (Supplementary Fig. 8d). Simultaneously, five consecutive cuts were made towards the olfactory bulb to remove the unwanted parts (Supplementary Fig. 8e). Note: the cut should be perpendicular to the rostrocaudal axis as this face will stick on the razor blade of the vibratome (Supplementary Fig. 8f). Brain slices containing SNr (Supplementary Fig. 8g) were selected. The 1-mm thick brain sections were placed in a type microscope alongside pre-cooled metal brackets (Supplementary Fig. 8h). Later, fine tweezers were used to remove the non-target nuclear mass tissues, and the SNr region was extracted as far as possible from the outside to avoid the SNc region (Supplementary Fig. 8i). The dissociated bilateral SNr nuclei (Supplementary Fig. 8j).

## Haematoxylin and Eosin (H&E) staining

The mice were euthanized by deeply anesthetized with pentobarbital sodium (100 mg/kg, i.p.) and then cervical dislocation, and liver tissue was removed for histology verification of acute liver injury. After immersed in 10% formalin, dehydrated with ethanol gradient, the liver tissues were subsequently permeabilized with xylene and paraffin-embedded. Sample blocks were cut into 4 μm thickness and stained with hematoxylin and eosin (H&E). Eventually, the slide was observed under a brightfield microscope (Olympus, Japan). H&E slides were taken from three mice in each group, and 4 slides were taken from each liver tissue.

## Determination of ammonia levels in blood and brain

The mice were anesthetized with pentobarbital sodium (100 mg/kg, i.p.), and the blood samples were obtained by removing the eyeball and sacrificed for determination of ammonia levels in blood. The whole brains were isolated and homogenized for determination of ammonia levels in brain. Blood samples and brain samples were centrifuged at 3500 g at 4 °C for 10 min and detected by using a commercial kit (Shuahua Biology, SH-9371, China). The steps were strictly performed following the instructions of the kits.

## Enzyme-linked immunosorbent assay (Elisa)

To observe indicators of changes in liver function, we measured the serum levels of alanine transaminase (ALT, C009-2-1), aspartate transaminase (AST, C010-2-1), and total bilirubin (TBil, C019-1-1), by using commercial kits (Nanjing Jiancheng Bioengineering Institute, China). Superoxide dismutase (SOD, A001-3-2), and glutathione peroxidase (GPX, A005-1-2) in peripheral blood were measured using commercial kits (Nanjing Jiancheng Bioengineering Institute, China).

## Mitochondrial Network Analysis (MiNA)

Mitochondrial Network Analysis (MiNA) uses the freely available FiJi distribution of the ImageJ platform and amalgamates open-source tools into a simple macro toolset. The brain sections of GAD2-Mito-GFP mice were used for MiNA analysis under the inverted laser scanning confocal microscope (Olympus FV1000). The number of the individual mitochondria was evaluated. Four mice were taken from each group, and at least seven slides were taken from each brain tissue.

## Determination of reactive oxygen species (ROS)

ROS fluorescent probe−(BBoxiProbe®O$^{13}$, BB-470513, China) staining was adopted to determine reactive oxygen species (ROS) in the SNr. BBoxiProbe®O$^{13}$, a cell-permeable oxidative-sensitive fluorescent dye, was oxidized to ethidium by superoxide, which, subsequently binded to DNA in the nucleus and emits red fluorescence. After the frozen brains were cut at a thickness of 15 μm by using a CM 3050 cryostat (Leica Microsystems), the sections were incubated with A liquid in PBS for 10 min at 37 °C in a humidified chamber protected from light. Then, the images were captured with a confocal fluorescence microscope (FV-1000) and positive cells in each section were quantified by using the Image J.

## Western blot

Proteins were isolated from homogenized tissues of bilateral SNr areas using modified RIPA buffer supplemented with halt protease and Phosphatase Inhibitor (Thermo Fisher Scientific). Protein concentrations were measured using Assay Reagent Kit. Protein assay was separated using 12.5% SDS-PAGE and transferred to a 4.5 um PVDF membrane, which was then blocked with 5% skim milk in TBST and incubated at room temperature for 2 h. The primary antibodies were Mouse anti-SOD1(1:1000, Cat. #A0274, Abclonal, CN), Mouse anti-GPX1(1:1000, Cat. #A11166, Abclonal), Rabbit anti-UCP2(1:1000, Cat. #89326, Cell Signaling Technology, UAS), Mouse anti-UCP4 (1:1000, Cat. #sc-365295, Santa Cruz Biotechnology, USA), Rabbit anti-UCP5 (1:1000, Cat. #A13731, Abclonal) for oxidative stress; Rabbit anti-LC3B (1:1000, Cat. #3868, Cell Signaling Technology), Rabbit anti-PINK1 (1:1000, Cat. #ab23707, Abcam, MA, UK) for autophagy; Rabbit anti-DRP1 (1:1000, Cat. #8570 s, Cell Signaling Technology), Phospho-Ser616 DRP1 (1:1000, Cat. #3455, Cell Signaling Technology), Rabbit anti-MFN2 (1:1000, Cat. #9482 S, Cell Signaling Technology), Rabbit anti-FIS1 (1:1000, Cat. #A19666, Abclonal), Rabbit anti-MFF (1:1000, Cat. #84580 s, Cell Signaling Technology) for mitochondrial function; and Mouse anti-β-ACTIN (1:5000, Cat. # AC004, Abclonal) as internal reference, overnight incubation at 4 °C. The study used horseradish peroxidase (HRP)-conjugated secondary antibodies HRP-Goat anti-Mouse (1:5000, Cat. #A21010 Abbkine) and HRP-Goat anti-Rabbit (1:5000, Cat. #A21020 Abbkine, CN). Membranes were scanned by the Fusion FX EDGE chemiluminescence imaging and gray value analysis was conducted using ImageJ.

## Stereotaxic AAV viral vectors injection

The viral vectors employed in this work included: pAAV-hSyn-DIO-mCherry, AAV8-hSyn-DIO-mCherry, AAV8-hSyn-DIO-hM4Di (Gi)-mCherry, AAV8-hSyn-DIO-hM3Dq (Gq)-mCherry, pAAV-DIO-mCherry-NC-WPRE (*Ucp2* NC), pAAV-DIO-mCherry-shRNA (*Ucp2*)-WPRE (*Ucp2* KD), and pAAV-DIO-mCherry-P2A-*Ucp2*-WPRE (*Ucp2* OE) were provided by OBiO (China). The mice were sedated with sodium pentobarbital (100 mg/kg, i.p) before being placed in stereotaxic apparatus (RWD, China). The mouse brain atlas was used to map the fiber tip placement (from bregma, anteroposterior: −3.4 mm; mediolateral: ±1.5 mm; dorsoventral: −4.55 mm). An injector (Hamilton Co., Reno, NV) was connected with a 5-μL Hamilton syringe. All mice were injected bilaterally with 1 μL viral vector ($1.0 \times 10^{12}$ viral genomes [VG] / mL) at a rate of 100 nL/min. After injection, the needle was left in situ for 10 min before being slowly withdrawn to avoid possible leakage from the needle track.

## Direct local high ammonia stimulation into SNr

$NH_4Cl$ (Sigma, Lot#BCCG5690) 1 μL was injected at a rate of 100 nL/ min into bilateral SNr at a concentration of 10 mM, with the same dose of saline serving as a control.

## c-FOS (+)- neuron labeling (TRAP)

According to the previous reports[52], TRAP2 mice received i.p. injection of tamoxifen (TM, Sigma, Lot#BCCD3910) for 5 consecutive days. TM was dissolved in corn oil (Sigma, Lot#66633) and its final concentration was 100 mg/mL. At the state of HE model or local high ammonia stimulation, the specifically activated neurons would become FOS-positive which could be reflected by red fluorescence. Then a confocal fluorescence microscope was used to analyze FOS+ neurons (FV-1000, Olympus, Japan).

## Designer receptors exclusively activated by designer drugs (DREADDs) manipulation

To accurately regulate the GAD2-expressing GABA population, viral vectors carrying Designer receptors exclusively activated by designer drugs (DREADDs) elements (Gi or Gq) were injected into the bilateral SNr areas of GAD2-ires-cre mice. The mice received i.p. injection of ligand clozapine N-oxide (CNO, Sigma, C0832) or Deschloroclozapine dihydrochloride (DCZ, MCE, HY-42110). CNO was dissolved in saline (0.9% NaCl) and its concentration was 2 mg/mL. DCZ was dissolved in saline and its concentration was 0.02 mg/mL. There were 3 mice in each group and at least 3 slices were taken from each brain tissue to observed virus injection site.

## Electrophysiological slice recordings

Electrophysiology recording was performed as described in previous research[46]. The mice were anesthetized with pentobarbital sodium (40 mg/kg, i.p.). In terms of obtaining SNr slices, GAD2-expressing GABA population in SNr (mCherry/Gi/Gq) were decapitated and the brains were dissected rapidly and placed in ice-cold oxygenated (95% $O_2$ and 5% $CO_2$) solution containing the following (in mM): NaCl 88, KCl 2.5, $MgCl_2$ 7, $CaCl_2$ 0.5, $NaH_2PO_4$ 1.25, $NaHCO_3$ 25, sucrose 75, pH 7.4. Brain slices with a thickness of 300 μm were cut with a vibrating microtome (Leica VT1200S, Germany) and placed in artificial cerebrospinal fluid (ACSF) containing the following concentrations (in mM): NaCl 126, KCl 2.5, $NaH_2PO_4$ 1.25, $NaH_2PO_4$ 1.25, $NaHCO_3$ 25, $CaCl_2$ 2, $MgSO_4$ 2, glucose 10. Whole-cell patch-clamp recording was performed by a patch-clamp amplifier (Axon 700B, MD, USA). The tip diameter of the electrode used for recording cells was ~2 μm, and the impedance was about 4-5 MΩ. For recording the spontaneous firing frequency of GAD2-expressing GABA population expressing mCherry fluorescence in SNr, the following perfusion electrode solutions were routinely administered (mM): K-gluconate 133, NaCl 8, EGTA 0.6, Mg ATP 2, $Na_3$•GTP 0.3, HEPES 10, The pH was adjusted to 7.2–7.4 with KOH and the osmolarity was 280–290 mOsm. The spontaneous action potential was stably recorded for 1 min before and after CNO (5 μM) application. The Clampfit 11.0.3 software was adopted to analyze the frequency of spontaneous neuron firing within 30 s.

## Tissue preparation and immunofluorescence analysis

The mice were anesthetized with pentobarbital sodium (100 mg/kg, i.p.) and transcardially perfused with 50 mL of 0.01 M phosphate buffer saline (PBS, pH 7.4) and thereafter fixed with 4% paraformaldehyde (pH 7.4). The brains were then collected and subsequently immersed in 0.1 M PB containing 30% sucrose at 4 °C. Coronal sections via the SNr area with a thickness of 30 μm were sectioned on a cryostat (Leica CM1800; Heidelberg, Germany). Sections were incubated in a blocking PBS buffer containing 0.3% Triton X-100 and 0.05% sodium azide, 10% bovine serum albumin for 30 min at room temperature. The primary antibodies included mouse anti- Mouse anti-GAD67 (GAD1) (1:100, Cat. #ab26116, Abcam), Rabbit anti-GAD65 (GAD2) (1:50, Cat. #5843 s, CST), Rabbit anti-Parvalbumin (1:200, Cat. #80561 s, CST), Rabbit anti-vGlut1 (1:500, Cat. #a12879, Abclonal), Rabbit anti-GFAP (1:200, Cat. #12389 s, CST), and Rabbit anti-NeuN (1:50, Cat. #ab190565, Abcam). The secondary antibodies Alexa 488-Goat anti-Mouse (1:500, Cat. #A23210, Abbkine) and Alexa 488-Goat anti-Rabbit (1:500, Cat. #A23220, Abbkine) were used, together adding DAPI (1:500, Cat. #BMD0063, Abbkine) at room temperature for 12 min. All images were taken under FV-1000 confocal fluorescence microscope. There were 3 mice in each group and at least 4 slices were taken from each brain tissue.

## Analysis of locomotor activities and coordination

Locomotor activities of mice were assessed automatically via Video Tracking in the Open Field (OF) test. In addition, the motor defect and coordination were analyzed by using the rotarod test and CatWalk analysis.

Open Field (OF) test: Half an hour after the CNO (Sigma, C0832) or 10 min after DCZ (MCE, HY-42110) injection, the mice were allowed to discover an open-field arena (45 cm × 45 cm × 45 cm) for 10 min. Mice were acclimated to the behavior room for at least 1 h before each test. Analysis of the animal's trajectory, including the animal's total distance, speed, activity number and activity time, was ensured.

Rotarod test: All mice were trained for three consecutive days before formal behavior testing. During the test, the mice were placed on Rotarod test device (BYZ-007, China) and started to rotate from 5 revolutions per minute (rpm), and increased to the maximum speed of 40 rpm within 150 s, for a total of 5 min. The time when the mouse fell was recorded and each mouse was tested 3 times.

CatWalk test: The total length of the CatWalk test used in this experiment was 130 cm, and the Walkway length and Walkway Width for detecting the movement of mice were 23.86 cm and 4.08 cm respectively. Each mouse was placed on the CatWalk XT (Noduls) and displaced after at least 3 runs which were evaluated by the Noduls software. Each paw identification was detected automatically by the software and manually checked to confirm the accuracy.

## Metabolic cages

The respiratory energy metabolism monitoring system (Coulumbus Oxymax/CLAMS) was employed to monitor the mice. Each mouse was placed individually in metabolic cage for 24 h to ensure adaptation. The metabolic data were continuously recorded for 48 h, and the stable 24 h data was selected as the final efficient data. Oxygen consumption ($VO_2$), $CO_2$ release ($VCO_2$), respiratory exchange rate (RER) and activity were detected. And a respiratory exchange rate (RER) was calculated as the $VCO_2/VO_2$.

## Body temperature

A portable intelligent digital thermometer (Middle way instrument, China) was used to measure the mice at room temperature. The probe of the detector was coated with lubricating oil before use and placed at the same depth each time, and the value on the detector was recorded after the temperature remained constant. The body temperature of mice was measured every 24 h, and continuously monitored for 7 days.

## Mitochondrial isolation

SNr mitochondria were isolated from the freshly dissected tissues of anesthetized with pentobarbital sodium (100 mg/kg, i.p.) mice using commercial mitochondrial extraction kits (Solarbio, SM0020, China). First, the SNr tissue was sectioned as small as possible using scissors, and then broken with pre-cooled TissueLyser (QIAGEN). Centrifugation was performed at 1000 g for 5 min at 4 °C for 2 times, then at 12,000 g for 10 min. The supernatant was discarded and the pellets containing mitochondrial proteins were rinsed twice. Lastly, mitochondrial proteins were quantified using the BCA method.

## Measurements of respiration of isolated mitochondria

Respiration was measured in freshly isolated mitochondria by monitoring oxygen consumption with a MitoXpress Xtra oxygen consumption assay probe (LUCXEL BIOSCIENCES) using the CLARIO star plus instrument (BMG LABTECH). The measuring buffer (250 mmol/L sucrose, 15 mmol/L KCl, 1 mmol/L EGTA, 5 mmol/L $MgCl_2$, 30 mmol/L $K_2HPO_4$, pH = 7.4) was diluted at 1:10 and 100 μL was added to each. The mitochondria were diluted to the target concentration by adding 50 μL. The final concentration of the substrate succinic acid (Sigma, S3674) was diluted to 25 mmol/L and the ADP (SLCK1130) was diluted to 1.65 mmol/L with the measuring buffer solution, and 50 μL of this solution was added. Exactly 100 μL mineral oil per well (preheat to 30 °C) was added to reduce interference from ambient oxygen. The fluorescence intensity of the Mpress Xtra probe was measured dynamically, and the program was set for mitochondrial respiration

monitoring for 60 min. The addition of ADP was recorded as state III respiration. Mitochondrial state IV respiration was observed without substrate consumption in the ADP group. The mitochondrial respiratory control ratio (RCR) is defined as state III/state IV.

## Enzymatic activity of the electron-transport-chain components III/IV

The complex activity was assessed with the mouse complex III (SH-M2032)/IV (SH-M0892) Elisa kit (SHUHUA BIOLOGY, China) according to the instructions of the manufacturer.

## ATP Measurement

An enhanced ATP assay kit (Beyotime, S0027, China) was used to measure total SNr tissue ATP levels according to the instructions of the manufacturer. Luminescence was measured using a luminometer at 560 nm. A standard curve was generated and used to calculate ATP concentration. BCA method was used to quantify the tissue protein.

## Brain water content

The liquid on the specimen surface was sucked with filter paper before calculating the wet weight of the whole brain (m1). The brain tissue was placed in a 120 °C oven for 48 h to achieve constant weight, and the total brain stem weight (m2) was obtained. Brain water content percentage was calculated by using the formula = $[(m1 - m2)/m1] \times 100\%$.

## Nissl staining

Mice were anesthetized with PBS via the heart and fixed with 4% paraformaldehyde. The brain was removed and maintained with 4% paraformaldehyde for 48 h before dehydrated with 75, 80, 95 and 100% alcohol and thereafter transparent with mice xylene, dipped in wax, and sliced into brain sections (5 μm thickness) with a microtome (Leica, Germany). After debenzenizing and dewaxing, the brain sections were stained in ziehl (Servicebio, China) for 2–5 min and then washed. Meanwhile, the brain slices were washed and baked. Lastly, the sections were observed under a microscope (Olympus, Japan) to determine the number of neurons in the SNr.

## Statistical analysis

In this paper, all data were represented as mean ± SD, with error bars representing SD. All statistical data were analyzed using the SPSS software version 26.0 and GraphPad Prism 8.02. For data with a normal distribution, paired or unpaired t-tests Student's t-tests were used to compare two groups. For more than two groups, statistical analysis was performed using one-way ANOVA, followed by the LSD post-analysis. $P < 0.05$ was considered statistically significant.

## Reporting summary

Further information on research design is available in the Nature Portfolio Reporting Summary linked to this article.

## Data availability

All data generated in this study are available upon request to the corresponding author. Source data are provided with this paper.

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

## Acknowledgements

We express our appreciation to Y.Z., translator currently working for Xi'an Aeronautic Polytechnic Institute. This project is funded by the following program: The General Programs of the National Natural Science Foundation of China 81870415 (Y.Y.); Youth Programs of the National Natural Science Foundation of China 82000551 (Y.B.); The joint construction project LHGJ20210804 (Y.B.). We thank all teachers of the Teaching Experiment Center of the Fourth Military Medical University and the Department of Hepatobiliary Surgery, Xi-Jing Hospital, Fourth Military Medical University for helpful insights and suggestions.

## Author contributions

Y.Y., Y.W., and Y.C., designed the research plan and interpreted experimental results. Y.B. performed all injection Virus, Elisa, mito-chondrial fission, autophagy, and oxidative stress experiments. K.L., X.L., and X.C., performed the identification of transgenic mice and behavior experiments. J.Z., performed drawing machine diagrams. F.W., J.C., Z.L, S.Z., and K.W. helped edit the manuscript.

## Competing interests

All authors declare no competing interests.
