## [Peer Review File · Nature Communications]

Effects of oxidative stress on hepatic encephalopathy pathogenesis in miceREVIEWER COMMENTS

Reviewer #1 (Remarks to the Author):

This study explores the role of mitochondrial dysfunction in driving MHE in a model of cirrhosis and suggests that using a mitochondria targeted approach to reduce oxidative stress, MHE could be resolved. My comments are as follows.

1. The authors justify their target of the SNr and GABA based on old data that have remained a hypothesis in the pathogenesis of HE. The rationale for the study and the hypothesis remains questionable. They need to explain.
2. HE is a disease condition that affects many functional domains with several clinical manifestations. I would like to see the authors replicate the data in different brain areas rather than only limited to the SNr.
3. There is a potential problem with interpreting the data obtained in TAA animals particularly those treated with the mitoquinone as there was concomitant improvement of liver function with the drug. I would suggest that the authors repeat the experiment in another clinically relevant model of advanced fibrosis and mHE such as bile duct ligation

Reviewer #2 (Remarks to the Author):

In this manuscript, Bai et al. mimicked acute hepatic encephalopathy (HE) in human using a mouse model via systemic injection of a liver toxin, thioacetamide (TAA) to induce acute liver injuries. These TAA-treated mice exhibited significant behavioral changes and differences in some standard markers of mitochondrial dynamics and oxidative stress in the brain. Then, the authors utilized FosCreERT2 strategy to further demonstrated pathogenic activation of SNr GABA neurons by TAA and ammonia, implicating that SNr GABA neurons may be involved in HE pathogenesis. Using chemogenetic approach (i.e. DREADDs), the authors were able to specifically activate or inhibit SNr Gad2-expression GABA neurons and that it modulated HE phenotypes. The authors also proposed two different approaches to ameliorate HE phenotypes induced by TAA, including overexpression of a mitochondrial uncoupling protein (UCP2) in the SNr, and systemic injection of an antioxidant, mitoquinone (MitoQ). Both approaches showed improvement in markers of mitochondrial dynamics and oxidative stress in TAA-treated mice.

Significant efforts have been made to demonstrate specific role of SNr GABA neurons in the pathogenesis of HE. This is a logical and interesting study which utilized different genetic manipulation approaches in mice to visualize and specifically activate/inhibit SNr GABA neurons. Whilst a significant portion of this paper presented differences in various mitochondrial stress markers and behavioral

changes in TAA-treated mice, these findings were not novel but were mostly reported in previous publication of the same author (Bai Y, et al., *The Anatomical Record*. 2019; 302:1169-1177). To justify the novelty of this manuscript, the authors should highlight the novel approaches to improve brain mitochondrial dynamics and functions under HE condition. Based on the data in the current form, there are major concerns regarding the conclusion of this study.

Major concerns:

1. This manuscript title emphasizes mitochondrial redox stress based treatment in HE. However, the current data is insufficient to support the conclusion only based on measurements of a few mitochondrial markers by Western blots. A comprehensive metabolic phenotyping in live animals (e.g. CLAMS™) after treatment may be necessary. This may involve basal metabolic rate of treated animals, yielding quantitative measurements of oxygen consumption (VO₂), carbon dioxide production (VCO₂), respiratory exchange ratio (RER; indicator of substrate utilization), total activity, and body temperature to demonstrate solid improvement over a certain period of time.
2. Fig. 3H: Patch clamp recordings showed that chemogenetically activated SNr neurons exhibited increased firing. This is by far the only direct evidence to show neuronal activation. However, the tracing appears that such effect was sustained only for a very short period time after CNO and lost. How such a short term effect can be translated into an observable behavioral changes in the Open Field test and RotaRod? Any proposed molecular mechanism that could explain why activated SNr neurons modulated motor behavioral deficits?
3. In the TAA-induced model, does HE condition affect neuronal cell viability in SNr or only the activity of the neurons was affected? This may be related to the prognosis of HE in human. Are these impairments in the brain reversible after recovery from liver damages? If so, would it be easier to address liver damages than the more complicated manipulation of a small sub-population of neurons in the brain?
4. Whilst most of the target markers in this study were claimed to be measured in SNr, how precise that only the SNr region of a mouse brain can be dissected without confounded by tissues from the adjacent brain regions?
5. The authors claimed that overexpressing UCP2 alleviated TAA-induced MHE by stabilizing mitochondrial functions. However, there is no direct experimental evidence to support such claim. Assays on mitochondrial respiration (i.e. State III/State IV respiration) and ATP synthesis may be useful to determine how the proposed approaches could have improved mitochondrial functions in the brain. Are there any differences in total ATP supply in SNr before and after relevant manipulation?

Minor concerns:

1. This study is limited to HE from acute liver injuries induced by TAA. Are there any wider implications to other HE conditions as a result of chronic liver injuries in human?
2. The authors only assessed UCP2 expression in the brain. Is there any contribution of other neuronal UCP isoforms expressing in the brain? Please justify and discuss.
3. The authors showed that MitoQ treatment significantly ameliorated hepatocyte necrosis induced by TAA and decreased ALT, AST and blood/brain ammonia. Obviously, the beneficial effects of MitoQ are on ameliorating liver injuries. It is crucial to differentiate whether MitoQ has any direct protective effects on SNr neurons, or the observed protective effects in the brain were secondary to liver recovery.

Can MitoQ pass through the blood-brain barrier?

4. Are there any side effects of UCP2 overexpression and MitoQ treatment in normal mouse brain without TAA treatment?

5. This study is purely pre-clinical. Are the brain markers of mitochondrial dynamics and oxidative stress in mice complementary to any clinical assessments in human HE?

6. Grammatical mistakes and typos are commonly found. Comprehensive revision by an English editor in the field is necessary.

Reviewer #3 (Remarks to the Author):

Comments for the manuscript : Mitochondrial redox stress based treatment of hepatic encephalopathy
NCOMMS-22-39565

The current study “Mitochondrial redox stress based treatment of hepatic encephalopathy” performed by Yunhu et al. provided original and significant data to the field of neuroscience, particularly the neuropathology of hepatic encephalopathy, through the assessment GABAergic neurons responsiveness in an animal model of minimal hepatic encephalopathy along with the role of ammonia per se in the oxidative stress-induced mitochondrial abnormalities in SNr.

Different technical approaches applied (molecular biology, immunohistochemical, electrophysiological, neurobehavioral, histological and biochemical tools) are sufficient to sustain the elaborated conclusions. A noteworthy results provided is : i) the role of ammonia as a key factor of GABAergic neurotransmission potentiating leading to brain inhibition in minimal HE. ii) The activation of Gad2 neurons and oxidative stress-induced mitochondrial abnormalities in SNr GABAergic neurons such finding make the current investigation of relevance to the understanding of HE neuropathology and therefore establishing new paths to the therapy.

However, the current study, as it is, presents some major issues:

In Introduction

1-Lin2: HE may arise even from extrahepatic complication (e.g congenital portosystemic venous shunts) so to complete the HE definition authors should take into consideration such data.

2-As the authors aimed in there manuscript to focus on the involvement of GABA in the neuropathology of minimal HE, the manuscript title should reflect that

3-authors provided a brief timeline of the evolution of GABA concept in the HE neuropathology, however, the neurosteroids theory with relevance to GABA tonus is of crucial importance and should be as well evoked.

In material and methods section:

1-The authors used hepatotoxic mice model induced HE (150 mg TAA/kg during 3 days). Actually, the HE

in TAA mice models is generally not well characterized (according to the recent 2021 ISHEN guideline for animal models of HE). As well, the used dose of TAA belongs to the interval of used doses to elicit acute HE (100-300mg/Kg) (according to the ISHEN guideline).

- How dose authors qualified there TAA mice as MHE model? The biochemical data provided (hepatic markers and ammonia) are not sufficient to sustain that idea.

- Dose the authors checked the brain edema in their TAA mice?

2- why the authors used the elevated plus maze in the context of assessing locomotor function? The EPM is more specific test to anxiety state assessment rather than locomotion, moreover, by checking the results, no data is provided with relevance to the EPM test!

3-No details are provided in regarding the static and dynamic parameters to be assessed using the Cat Walk test.

4-no indication regarding the volume and the administration route (i.p or s.c) of the dextrose, KCl and NaCl solution; the volume of the administrated solution depends to the injection route.

5-Remove "material and method" from page 8

Under results section:

1-the authors should provide an additional photomicrograph of a coronal section through the targeted brain area showing the site of intracerebral injection.

2-The HE stained liver slices showed in fig 1, 5 and 7 should be well presented to highlight the histological lesions (high magnification)

3-Captions for the last 6 figures (fig 7, 8, 9, 10, 11 and 12) are missing along with their titles.

2-some of the presented parameters of Open field test in the results section are not mentioned in the material and methods sections (e.g main speed)

3-As well, the cited OF parameters (the distance traveled to the center, and the number of times the animal entered the center) are not shown in the results section

4-In Fig1 and Fig 3, the titles should be shortened.

El Hiba Omar

Reviewer #4 (Remarks to the Author):

In the current manuscript by Bai et.al., authors try to explore the molecular mechanism by which increased GABAergic tone in SNr causes bradykinesia in hepatic encephalopathy. The authors used the DREADD and TRAP2 strategies to determine that oxidative stress and mitochondrial abnormalities are involved in HE. The manuscript also shows that Gad2 expressing GABA neurons within the SNr are important contributors to locomotor injury in the HE situation. Overall, the current manuscript authors used multiple approaches to address the role of GABAergic tone in SNr in HE. Below are some of the queries I feel need to be addressed.

Major:

The authors took advantage of the DREADD system to explore the molecular mechanism. The authors looked at UCP2 expression upon Gi and Gq DREADD activation. However, authors should also look at the expression profile of other autophagy and oxidative stress genes upon DREADD activation or inhibition to rule out the possibility of any other mechanisms.

Earlier reports showed that hyperammonemia induces activation of astrocytes and microglia, so authors should discuss the possibility of cross-talk between the neurons and other cell types.

The authors used 2 mg/kg of CNO, and earlier studies mostly used 1 mg/kg or less, as CNO can be metabolized and can have off-target effects. To rule out this possibility, authors should consider using DCZ in at least one or two major experiments to confirm the same phenotype observed.

The authors should use some pharmacological interventions to validate Fig. 6. In addition to ammonia toxicity, authors need to at least discuss the possibility of neuronal crosstalk between the CNS and peripheral tissues, which may also contribute to the claimed observations.

Minor:

The current study definitely helps us understand the role of SNr Fad2-expressing neurons in HE, but claiming a new target is far-fetched. Maybe the language can be toned down.

It would also be nice if the authors considered describing a few sentences about the limitations of the study.

UCP2 KD seems to be around 50–60%, and this KD is achieving complete loss of phenotype. Can the authors confirm the percentage of KD they observed?

In the second line of the introduction, I believe it is acute or chronic liver failure.

In the methods section, "Determination of ammonia levels in blood and brain," the authors talked about only blood; I need some clarification on this. Also used term killed, I'm not sure this is an appropriate term to use.

Response to Reviewer #1

REVIEWER COMMENTS

Reviewer #1 (Remarks to the Author):

This study explores the role of mitochondrial dysfunction in driving MHE in a model of cirrhosis and suggests that using a mitochondria targeted approach to reduce oxidative stress, MHE could be resolved. My comments are as follows.

1. The authors justify their target of the SNr and GABA based on old data that have remained a hypothesis in the pathogenesis of HE. The rationale for the study and the hypothesis remains questionable. They need to explain.

[Response]

Dear reviewers, thank you for your constructive suggestions regarding our manuscript.

Reports indicate that in HE situations, motor activity is modulated via different neuronal circuits than in normal statements ^[1]. At the normal statement, neuronal circuits involved in motor activity include the nucleus accumbens → ventral pallidum → mediodorsal thalamus → prefrontal cortex ^[2]. In the HE situation, the neuronal circuits involved in hypokinesia include the nucleus accumbens → SNr → ventromedial thalamus and prefrontal cortex. Moreover, scholars have shown that targeting inhibition the Gad2 neuron of SNr can activity significantly increased motor initiation and reduced motor termination ^[3]. Therefore, SNr is critical for locomotor dysfunction in HE. The current research has indicated the hyperammonemia level could directly activate GABAergic neurons within the SNr to express activation marker Fos. We think these activated neurons may represent the hyperactivation state. There are two reasons to explain. First, we have found that the chemogenetic inhibition of such population exerts the beneficial effect on the bradykinesia disorder in MHE mice. On the contrary, the chemogenetic activation of such population exerts the worse effect on the bradykinesia disorder in MHE mice. The above results were further verified in DCZ. DCZ is a metabolic product of CNO and has a stronger effect than that of CNO. Above phenomenon might confirm the theory of the increased GABA tone in pathological mechanisms underlying HE.

Reference:

[1] Cauli O, Llansola M, Erceg S, et al. Hypolocomotion in rats with chronic liver

failure is due to increased glutamate and activation of metabotropic glutamate receptors in substantia nigra. *J Hepatol.* 2006 Nov;45(5):654-61.

[2] Galaj E, Han X, Shen H, et al. Dissecting the Role of GABA Neurons in the VTA versus SNr in Opioid Reward. *J Neurosci.* 2020; 11;40(46):8853-8869.

[3] Liu D, Li W, Ma C, et al. A common hub for sleep and motor control in the substantia nigra. *Science.* 2020. 24; 367(6476):440-445.

2. HE is a disease condition that affects many functional domains with several clinical manifestations. I would like to see the authors replicate the data in different brain areas rather than only limited to the SNr.

[Response]

Thank you for your advice.

Hepatic encephalopathy has been shown to result in the changes in the neuronal circuits between the basal ganglia and the cerebral cortex. Therefore, we have further added the experiments to observe whether TAA could induce the changes of the expression levels of autophagy markers (LC3B and Pink1), antioxidative stress markers (SOD1 and GPx1), mitochondrial uncoupling proteins (Ucp2, Ucp4, and Ucp5), and mitochondrial factors including pro-mitochondrial fission factors of dynamin-related protein 1 (Drp1) and its phosphorylated version of p-Drp1, the mitochondrial fission factor (Mff) in charge of pro-midzone-type fission, the mitochondrial fission 1 (Fis1) in charge of pro-peripheral-type fission, and pro-mitochondrial fusion factor mitochondrial fusion 2 (Mfn2) relevant changes, in the cerebral cortex. Western blot results have showed that, in cerebral cortex, TAA could only induce the significantly increased levels of LC3B and Pink1, as well as the significantly decreased levels of SOD1 and GPx1. These data have been different from those in SNr. We deduce that it is possible that the SNr, but not cerebral cortex, is more sensitive to TAA-induced MHE and associated oxidative stress.

Please check these in the second part of the section of Results and associated Figs. 2J - P.

In addition, we have performed the experiments to observe the effect of Mito-Q and Ucp2OE on mitochondrial respiration in SNr of MHE mice. We have analyzed VO₂, VCO₂, RER, and Y-total activity during the dark, during the light, and total test period in Sham, TAA + Saline, and TAA + Mito-Q groups. We have found Mito Q could alleviate TAA-induced hypothermia in mice. The changes of mitochondrial State III

and State IV respiration, as well as mitochondrial average State III and State IV respiration, have been detected. Mitochondrial respiratory control rate (RCR), mitochondrial respiratory chain complex III and IV, and total ATP in SNr have been measured. The effect of MitoQ on basal metabolic rate in mice has been added.

These data are shown in Fig. 8 and Supplementary Fig. 7. Please check them.

3. There is a potential problem with interpreting the data obtained in TAA animals particularly those treated with the mitoquinone as there was concomitant improvement of liver function with the drug. I would suggest that the authors repeat the experiment in another clinically relevant model of advanced fibrosis and mHE such as bile duct ligation.

[Response]

Thank you very much for your comments.

According to your comments, we have performed the experiments on the MHE model with bile duct ligation (BDL).

Bile duct ligation (BDL) model has been described. Sodium pentobarbital (100 mg/kg, i.p) was used as anesthesia for the mice (10 - 12 w) during the BDL surgery. After skin preparation, the skin was sterilized using 70% ethanol, and an incision of about 1.5 cm was made along the ventral median line. The bile duct was separated from the portal vein and hepatic artery and dissociated for a certain length after opening the abdomen. The 5 - 0 suture was knotted at the upper end of the common bile duct. The bile duct at the upper margin of the pancreas was ligated again to ensure that it was effectively blocked without being cut off. The abdominal organs were rinsed using 0.9% NaCl solution and put back in situ. The abdominal cavity was then sutured and disinfected. This section of the experiment was divided into four groups: Sham operation was performed in the sham group, that is, free bile duct alone; Sham + Mito-Q group was given an intraperitoneal injection of Mito-Q (5 mg/kg, once every other day) after sham operation; BDL operation group and BDL + Mito-Q group, all mice were observed until the third week.

Please check this part in the section of Methods.

In the present study, we have used Mito-Q Consistent with previous reports^[26, 27], BDL induced mild hepatocytes necrosis (Fig. 4B), and increased the levels of ALT (Fig. 4C), AST (Fig. 4D), TBil (Fig. 4E), and blood ammonia (Fig. 4F). BDL mice showed the balance impairment by rotarod test (Fig. 4G) and the bradykinesia reflecting as the

decreased total distance, average speed, and the number of activities by open field test (Fig. 4H and I). We observed the similar deficiency of locomotor activity in BDL mice group to 150 mg / kg TAA group. It was shown that Mito-Q treatment alleviated BDL-induced hepatic injury (Fig. 4B - F) and movement disorder (Fig. 4G - I), which was similar to the results from TAA mice. Western blot results (Fig. 4J - O) showed at the first time that, in SNr, BDL induced the increase of Ucp2, LC3B, Pink1, pDrp1, Mff and Fis1, as well as the decrease of SOD1 and Mfn2. These results confirmed the oxidative stress and mitochondrial dynamic injury within the SNr at the statement of BDL-induced MHE. In addition, BDL mice receiving Mito-Q treatment were protected from BDL-induced oxidative stress, autophagy attack, and mitochondrial fragmentation (Fig. 4J - O). These results confirmed the beneficial effect of Mito-Q on liver damage and neuronal imbalance in SNr.

We have added the contents into the section of Result.

Please check it at the fourth paragraph.

Response to Reviewer #2

Reviewer #2 (Remarks to the Author):

In this manuscript, Bai et al. mimicked acute hepatic encephalopathy (HE) in human using a mouse model via systemic injection of a liver toxin, thioacetamide (TAA) to induce acute liver injuries. These TAA-treated mice exhibited significant behavioral changes and differences in some standard markers of mitochondrial dynamics and oxidative stress in the brain. Then, the authors utilized FosCreERT2 strategy to further demonstrated pathogenic activation of SNr GABA neurons by TAA and ammonia, implicating that SNr GABA neurons may be involved in HE pathogenesis. Using chemogenetic approach (i.e. DREADDs), the authors were able to specifically activate or inhibit SNr Gad2-expression GABA neurons and that it modulated HE phenotypes. The authors also proposed two different approaches to ameliorate HE phenotypes induced by TAA, including overexpression of a mitochondrial uncoupling protein (UCP2) in the SNr, and systemic injection of an antioxidant, mitoquinone (MitoQ). Both approaches showed improvement in markers of mitochondrial dynamics and oxidative stress in TAA-treated mice.

Significant efforts have been made to demonstrate specific role of SNr GABA neurons in the pathogenesis of HE. This is a logical and interesting study which utilized different genetic manipulation approaches in mice to visualize and specifically activate/inhibit SNr GABA neurons. Whilst a significant portion of this paper presented differences in various mitochondrial stress markers and behavioral changes in TAA-treated mice, these findings were not novel but were mostly reported in previous publication of the same author (Bai Y, et al., *The Anatomical Record*. 2019; 302:1169-1177). To justify the novelty of this manuscript, the authors should highlight the novel approaches to improve brain mitochondrial dynamics and functions under HE condition. Based on the data in the current form, there are major concerns regarding the conclusion of this study.

Major concerns:

1. This manuscript title emphasizes mitochondrial redox stress based treatment in HE. However, the current data is insufficient to support the conclusion only based on measurements of a few mitochondrial markers by Western blots. A comprehensive metabolic phenotyping in live animals (e.g. CLAMS™) after treatment may be necessary. This may involve basal metabolic rate of treated animals, yielding

quantitative measurements of oxygen consumption (VO₂), carbon dioxide production (VCO₂), respiratory exchange ratio (RER; indicator of substrate utilization), total activity, and body temperature to demonstrate solid improvement over a certain period of time.

[Response]

Dear reviewers, thank you for your careful review and constructive suggestions regarding our manuscript.

We have performed the experiments to observe the effect of Mito-Q on metabolic phenotyping in live animals of MHE mice. We have analyzed VO₂, VCO₂, RER, and Y-total activity during the dark, during the light, and total test period in Sham, TAA + Saline, and TAA + Mito-Q groups. We have found Mito-Q could alleviate TAA-induced hypothermia in mice.

These data are shown in Fig. 8 and Supplementary Fig. 7. Please check them.

2. Fig. 3H: Patch clamp recordings showed that chemogenetically activated SNr neurons exhibited increased firing. This is by far the only direct evidence to show neuronal activation. However, the tracing appears that such effect was sustained only for a very short period time after CNO and lost. How such a short term effect can be translated into an observable behavioral changes in the Open Field test and RotaRod? Any proposed molecular mechanism that could explain why activated SNr neurons modulated motor behavioral deficits?

[Response]

Thank you very much for your question.

In order to detect the activation of Gad2-expressing population within the SNr area, we have used Fos-cre ERT2 (TRAP2) mice which are designed to express tamoxifen (TM) - inducible, enhanced Cre recombinase (icre/ERT2) from the Fos promoter/enhancer elements, without interfering with the expression of endogenous Fos. In the present study, we have injected viral vectors containing double floxed sites (DIO) (AAV8-hSyn-DIO-mCherry) into bilateral SNr of TRAP2 mice. After treatment of CNO ^[1], we would observe Fos+ neurons expressing mCherry fluorescence in the SNr. These red neurons have represented the activated neuronal cells by MHE model or by direct ammonia injection into SNr. This part has been described in section of results. In addition, it has been reported that CNO takes effect about 30-40 minutes after injection, and the metabolic cycle is long, with a peak lasting about 2 h. Open Field test

and Rotarod can be completed within 2 hours ^[1]. We observed significant activation or inhibition of neurons after the administration of CNO in electrophysiology (replaced typical image). In order to further illustrate the role of chemogenetics, we used the metabolite of CNO, DCZ (faster acting and stronger acting), to conduct behavioral verification again. See the results in the Supplement Fig. 6 more details.

We think the molecular changes might be complicated. Recent research has shown that GABA neuron activation in the SNr could decrease open field movement while GABA neuron inhibition could increase activity in mice ^[2]. The 2020 Science study has further supported the idea that Gad2 neurons in the SNr are primarily responsible for controlling motor behavior ^[3]. In our previous studies, TAA could cause KCC2 (K⁺-Cl⁻ co-transporter) translocation from the neuronal membrane to the neuronal cytoplasm in mouse SNr-GABA neurons ^[4]. The present study has further observed the involvement of the molecular related to oxidative stress, autophagy injury and mitochondrial dysfunction in the activation of SNr^{Gad2} neurons during MHE statement. But the exact molecular mechanisms are not clear here.

Reference:

- [1] Guettier JM, Gautam D, Scarselli M, et al. A chemical-genetic approach to study G protein regulation of beta cell function in vivo[J]. Proc Natl Acad Sci USA. 2009; 106 (45): 19197-19202.
- [2] Galaj E, Han X, Shen H, et al. Dissecting the Role of GABA Neurons in the VTA versus SNr in Opioid Reward. J Neurosci. 2020. 11;40(46):8853-8869.
- [3] Liu D, Li W, Ma C, et al. A common hub for sleep and motor control in the substantia nigra. Science. 2020; 24;367(6476):440-445.
- [4] Yang YL, Li JJ, Ji R, et al. Abnormal chloride homeostasis in the substantia nigra pars reticulata contributes to locomotor deficiency in a model of acute liver injury. PLoS One. 2013; 8(5): e65194.

3. In the TAA-induced model, does HE condition affect neuronal cell viability in SNr or only the activity of the neurons was affected? This may be related to the prognosis of HE in human. Are these impairments in the brain reversible after recovery from liver damages? If so, would it be easier to address liver damages than the more complicated manipulation of a small sub-population of neurons in the brain?

[Response]

Thank you for your suggestion. To explore the effect of MHE on the neuronal cell

viability, Nissl staining was performed and the neurons of SNr were counted (see Supplementary Fig. 4). From our current results, it appears that the neuronal cell activity is induced by HE, but no influence- on viability, which is consistent with reversibility in clinical HE patients.

According to literature and clinical case-reports, symptomatic therapy and liver transplantation can improve encephalopathy status in patients with advanced liver disease. In this study, we found that the systemic administration of Mito-Q had a protective effect on both peripheral liver tissue and the central nervous system. This study aims to clarify that: firstly, antioxidant stress has an important significance in the treatment of HE both peripheral and central; Secondly, hepatic encephalopathy movement disorder is related to oxidative stress of GABA neurons in SNr. The last and most significant, our research reveals the possible target nuclei and target molecules of HE dyskinesia, and more fully understands the central mechanism.

4. Whilst most of the target markers in this study were claimed to be measured in SNr, how precise that only the SNr region of a mouse brain can be dissected without confounded by tissues from the adjacent brain regions?

[Response]

Thank you very much for this suggestion, which is really important for this study. Since accurate extraction of SNr is very important for subsequent relevant results, we try our best to be precise when extracting SNr nuclei (www.bio-protocol.org). In short, we used mouse precision meningeal (1 mm) to cut brain slices and then used fine tweezers to dissociate the target nuclei of interest under a microscope. Because SNr is dark color, it is easily distinguishable from peripheral nuclei. In order to avoid the mixing of SNc, we tried to discard part of the medial nucleus mass during separation. See detailed sampling steps and process pictures in Supplementary Fig. 8.

5. The authors claimed that overexpressing UCP2 alleviated TAA-induced MHE by stabilizing mitochondrial functions. However, there is no direct experimental evidence to support such claim. Assays on mitochondrial respiration (i.e. State III/State IV respiration) and ATP synthesis may be useful to determine how the proposed approaches could have improved mitochondrial functions in the brain. Are there any differences in total ATP supply in SNr before and after relevant manipulation?

[Response]

Thank you for your careful review and constructive suggestions regarding our manuscript.

To observe the effects of Ucp2OE and Mito-Q on mitochondrial function in SNr of MHE mice, mitochondrial respiration, electron respiratory chain complex III / IV, and ATP in total SNr were tested. See the results in the last (Fig.8) section for more details.

Minor concerns:

1. This study is limited to HE from acute liver injuries induced by TAA. Are there any wider implications to other HE conditions as a result of chronic liver injuries in human?

[Response]

Recent clinical reports have confirmed that oxidative accompanies hyperammonemia in HE patients and early reduction of oxidative stress in HE could contribute to minimize the neurotoxicity into severe outcomes. Montes-Cortes et al. ^[1] have compared the level of the oxidative marker malondialdehyde (MDA) in hepatic encephalopathy patients with chronic liver disease and in healthy volunteers. And the results have confirmed that the hyperammonemia in HE has been accompanied with the increased MDA. Importantly, they have reported that the early reduction of oxidative stress in HE could contribute to minimize the neurotoxicity into liver disease. Sfarti et al. ^[2] have performed a prospective case-control study which has included 40 patients divided into two groups: group A consisted of 20 cirrhotic patients with HE and increased systemic ammoniemia, and group B consisted of 20 cirrhotic patients with HE and normal systemic ammoniemia. The control group consisted of 21 healthy subjects matched by age and sex. The activity of SOD, GPx, MDA, and ammoniemia have been evaluated. They have found a significant decrease in SOD and GPx activity and also a significant increase of MDA levels in cirrhotic patients with HE and there is a significant correlation between the main oxidative stress markers and the levels of systemic ammonia. Görg et al. ^[3] have observed the significantly increased levels of other two oxidative markers of heme oxygenase (HO)1 and NADPH oxidase 4 (Nox4) in post-mortem brain samples from HE patients. All these clinical data, combined with the present pre-clinical results, have emphasized the influence of oxidative stress in HE pathogenesis in cirrhotic patients.

Reference:

- [1] Montes-Cortes DH, Olivares-Corichi IM, Rosas-Barrientos JV, et al. Characterization of oxidative stress and ammonia according to the different grades of hepatic encephalopathy. *Dig Dis*. 2020; 38(3):240-250.
- [2] Sfarti C, Ciobica A, Balmus IM, et al. Systemic oxidative stress markers in cirrhotic patients with hepatic encephalopathy: possible connections with systemic ammoniemia. *Medicina (Kaunas)*. 2020; 56(4):196.
- [3] Görg B, Karababa A, Schütz E, et al. O-GlcNAcylation-dependent upregulation of HO1 triggers ammonia-induced oxidative stress and senescence in hepatic encephalopathy. *J Hepatol*. 2019; 71(5):930-941.

2. The authors only assessed UCP2 expression in the brain. Is there any contribution of other neuronal UCP isoforms expressing in the brain? Please justify and discuss.

[Response]

Thank you very much for your valuable advice. According to previous literature reports, the UCP family consists of UCP1, UCP2, UCP3, UCP4 and UCP5. UCP2, UCP4 and UCP5 are expressed in the central nervous system. In the TAA-induced MHE model, we used western blot observation show that UCP4 and UCP5 were expressed in SNr and cortex, but no significant differences. See Fig. 2 for details.

3. The authors showed that MitoQ treatment significantly ameliorated hepatocyte necrosis induced by TAA and decreased ALT, AST and blood/brain ammonia. Obviously, the beneficial effects of MitoQ are on ameliorating liver injuries. It is crucial to differentiate whether MitoQ has any direct protective effects on SNr neurons, or the observed protective effects in the brain were secondary to liver recovery. Can MitoQ pass through the blood-brain barrier?

[Response]

In the present study, we have used Mito-Q as a mitochondria-targeted antioxidant drug. Mito-Q has been shown to target to accumulate in the matrix facing the surface of the mitochondrial inner membrane where it is optimally positioned to reduce mitochondrial-resourced ROS. Mito-Q is important for central nervous system because it could rapidly cross through the blood-brain barrier and accumulate in the brain ^[1-3]. In fact, Mito-Q has been used safely in Phase II clinical trials for Parkinson's disease patients and severe liver injury patients ^[4-5]. In the present research, we have revealed

that Mito-Q supplementation could improve liver damage, metabolism impairment, and mitochondrial dysfunction induced by TAA or BDL. Moreover, the targeted overexpression of Ucp2 has further shown the beneficial effect on the rescue of liver damage, metabolism impairment, and mitochondrial dysfunction induced by TAA. These results have indicated the key role of the communication of the central nervous system and peripheral tissues in different types of encephalopathy resulting from the injuries of liver, gut, kidney, and so on. The remission of the injuries of peripheral tissues could promote the recovery of neuronal function in the brain, at the same time, neuronal homeostasis contributes to the functional maintenance of peripheral tissues. We should pay more attention to the reciprocal effect of neurons and hepatic cells in the clinical treatment of HE.

Reference:

- [1] Fields M, Marcuzzi A, Gonelli A, et al. Mitochondria-targeted antioxidants, an innovative class of antioxidant compounds for neurodegenerative diseases: perspectives and limitations. *Int J Mol Sci.* 2023; 24(4):3739.
- [2] Tabet M, El-Kurdi M, Haidar MA, et al. Mitoquinone supplementation alleviates oxidative stress and pathologic outcomes following repetitive mild traumatic brain injury at a chronic time point. *Exp Neurol.* 2022; 351:113987.
- [3] Chen W, Guo C, Huang S, et al. MitoQ attenuates brain damage by polarizing microglia towards the M2 phenotype through inhibition of the NLRP3 inflammasome after ICH. *Pharmacol Res.* 2020; 161:105122.
- [4] E.J. Gane, F. Weilert, D.W. Orr, et al. The mitochondria-targeted anti-oxidant mitoquinone decreases liver damage in a phase II study of hepatitis C patients. *Liver Int.* 2010; (30) 1019-26.
- [5] B.J. Snow, F.L. Rolfe, M.M. Lockhart, et al. Protect Study, A double-blind, placebo-controlled study to assess the mitochondria-targeted antioxidant MitoQ as a disease-modifying therapy in Parkinson's disease. *Mov Disord.* 2010, (25) 1670-

4. Are there any side effects of UCP2 overexpression and MitoQ treatment in normal mouse brain without TAA treatment?

[Response]

Thank you for your advice. In order to observe whether overexpression of Ucp2 and injection of Mito-Q have side effects on normal mice, we conducted corresponding

treatment in normal mice, detected mouse behavior, collected blood samples for ALT and AST, and performed liver H&E pathological staining. Compared with untreated control mice, there was no significant statistical difference in the above indexes. See Supplementary Fig.3 materials for details.

5. This study is purely pre-clinical. Are the brain markers of mitochondrial dynamics and oxidative stress in mice complementary to any clinical assessments in human HE?

[Response]

As described above, several evidences have confirmed the important role of oxidative stress in HE. For example, Montes-Cortes et al. ^[1] have confirmed that the hyperammonemia in HE has been accompanied with the increased MDA. Sfarti et al. ^[2] have found a significant decrease in SOD and GPx activity and also a significant increase of MDA levels in cirrhotic patients with HE. Görg et al. ^[3] have observed the significantly increased levels of (HO)1 and Nox4 in post-mortem brain samples from HE patients. All these clinical data, combined with the present pre-clinical results, have emphasized the influence of oxidative stress in HE pathogenesis in cirrhotic patients.

Reference:

- [1] Montes-Cortes DH, Olivares-Corichi IM, Rosas-Barrientos JV, et al. Characterization of oxidative stress and ammonia according to the different grades of hepatic encephalopathy. *Dig Dis.* 2020; 38(3):240-250.
- [2] Sfarti C, Ciobica A, Balmus IM, et al. Systemic oxidative stress markers in cirrhotic patients with hepatic encephalopathy: possible connections with systemic ammoniemia. *Medicina (Kaunas).* 2020; 56(4):196.
- [3] Görg B, Karababa A, Schütz E, et al. O-GlcNAcylation-dependent upregulation of HO1 triggers ammonia-induced oxidative stress and senescence in hepatic encephalopathy. *J Hepatol.* 2019; 71(5):930-941.

6. Grammatical mistakes and typos are commonly found. Comprehensive revision by an English editor in the field is necessary.

[Response]

Thank you very much!

And we have asked a professional editor company, Zibo Yimore Translation CO. LTD, China to provide the English proofreading services.

Response to Reviewer #3

Reviewer #3 (Remarks to the Author):

Comments for the manuscript: Mitochondrial redox stress based treatment of hepatic encephalopathy NCOMMS-22-39565

The current study “Mitochondrial redox stress based treatment of hepatic encephalopathy” performed by Yunhu et al. provided original and significant data to the field of neuroscience, particularly the neuropathology of hepatic encephalopathy, through the assessment GABAergic neurons responsiveness in an animal model of minimal hepatic encephalopathy along with the role of ammonia per se in the oxidative stress-induced mitochondrial abnormalities in SNr.

Different technical approaches applied (molecular biology, immunohistochemical, electrophysiological, neurobehavioral, histological and biochemical tools) are sufficient to sustain the elaborated conclusions.

A noteworthy results provided is : i) the role of ammonia as a key factor of GABAergic neurotransmission potentiating leading to brain inhibition in minimal HE. ii) The activation of Gad2 neurons and oxidative stress-induced mitochondrial abnormalities in SNr GABAergic neurons such finding make the current investigation of relevance to the understanding of HE neuropathology and therefore establishing new paths to the therapy.

However, the current study, as it is, presents some major issues:

In Introduction

Lin2: HE may arise even from extrahepatic complication (e.g congenital portosystemic venous shunts) so to complete the HE definition authors should take into consideration such data.

[Response]

Thank you for your valuable advice. HE is classified into three categories according to etiology, and this study is mainly aimed at studying the mechanism related to HE induced by liver injury. In this study, a model induced chronic liver injury (BDL) was added and verified to further refining the definition of HE. See Section 4 of the results for details.

2-As the authors aimed in there manuscript to focus on the involvement of GABA

in the neuropathology of minimal HE, the manuscript title should reflect that.

[Response]

Thank you for your advice. In order to make the title more accurately reflect the main idea of the article, we have renamed the title as “Mitochondrial redox stress based treatment of minimal hepatic encephalopathy”.

3-authors provided a brief timeline of the evolution of GABA concept in the HE neuropathology, however, the neurosteroids theory with relevance to GABA tonus is of crucial importance and should be as well evoked.

[Response]

Thank you for your advice. We have added to the preface the hypothesis of neurosteroid theory related to GABA tension in HE. See the third paragraph of the introduction.

In material and methods section:

1-The authors used hepatotoxic mice model induced HE (150 mg TAA/kg during 3 days). Actually, the HE in TAA mice models is generally not well characterized (according to the recent 2021 ISHEN guideline for animal models of HE). As well, the used dose of TAA belongs to the interval of used doses to elicit acute HE (100-300mg/Kg) (according to the ISHEN guideline).

[Response]

Thank you very much for your comments.

We have the following four reasons to use TAA mice model in the present study in the original manuscript. First, in the 2009 and 2021 ISHEN guidelines, the TAA model has been recommended for rats. These guidelines have not excluded the applicability of TAA on mice. Second, there is a literature review to indicate that TAA-induced symptoms in mice are similar to that of human acute hepatic liver disease ^[1]. Third, previous reports have shown the significant dysfunction of central nervous system in TAA mice ^[2-3]. Fourth, our research group has studied the mice treated with TAA. We have confirmed the increased blood ammonia level, the hepatic injury and the bradykinesia dysfunction in TAA mice. And the associated data have been published ^[4-5].

But we think the comments of the respected reviewer are worthy elucidating. So, in the present manuscript, we have added the following data. We have screened the

effects of three different concentrations of 100 mg / kg, 150 mg / kg, and 200 mg / kg of TAA on mice. And we have found that the TAA with the concentration of 150 mg / kg could induce stable hepatic injury and bradykinesia performance in mice. The following experiments including DREADDs manipulation and Mito-Q treatment have been all based on the mice receiving this TAA concentration. The associated data have been provided in Supplementary Fig.1.

In addition, another MHE model induced by chronic liver injury (BDL) has been added. The associated results have been provided in Fig. 4. We have studied the same parameters on BDL mice, including the hepatic enzymes and H&E staining, blood ammonia level, locomotor activities, and the protein expression levels of SOD, GPx, LC3, Pink1, Drp1, pDrp1, Mfn2, MFF, Fis1. All these results have been compared with those from TAA mice. So in the present manuscript, the conclusion from TAA model and BDL model is consistent.

- [1] Sun X, Han R, Cheng T, et al. Corticosterone-mediated microglia activation affects dendritic spine plasticity and motor learning functions in minimal hepatic encephalopathy. *Brain Behav Immun.* 2019; 82: 178-187.
- [2] Cheng L, Wang X, Ma X, et al. Effect of dihydromyricetin on hepatic encephalopathy associated with acute hepatic failure in mice. *Pharm Biol.* 2021; 59(1): 557-564.
- [3] Lou P, Shen Y, Zhuge A, et al. Dose-Dependent Relationship between Protection of Thioacetamide-Induced Acute Liver Injury and Hyperammonemia and Concentration of *Lactobacillus salivarius* Li01 in Mice. *Microbiol Spectr.* 2021; 22;9(3): e0184721.
- [4] Bai Y, Bai Y, Wang S, et al. Targeted upregulation of uncoupling protein 2 within the basal ganglia output structure ameliorates dyskinesia after severe liver failure. *Free Radic Biol Med.* 2018; 124: 40-50.
- [5] Yang YL, Li JJ, Ji R, et al. Abnormal chloride homeostasis in the substantia nigra pars reticulata contributes to locomotor deficiency in a model of acute liver injury. *PLoS One.* 2013; 8(5): e65194.

How dose authors qualified there TAA mice as MHE model? The biochemical data provided (hepatic markers and ammonia) are not sufficient to sustain that idea.

[Response]

Thank you very much for your comments.

We have screened the effects of three different concentrations of 100 mg / kg, 150 mg / kg, and 200 mg / kg of TAA on mice. And we have found that the TAA with the concentration of 150 mg / kg could induce stable hepatic injury and bradykinesia performance in mice. The associated data have been provided in Supplementary Fig.1. In addition, another commonly MHE model induced by chronic liver injury (BDL) has been added. The associated results have been provided in Fig. 4. All these results have been compared with those from TAA mice. So in the present manuscript, the conclusion from TAA model and BDL model is consistent.

Dose the authors checked the brain edema in their TAA mice?

[Response]

Thank you very much for your valuable advice. Hepatic encephalopathy induced by acute liver failure is indeed mostly considered to be related to cerebral edema in current studies. Therefore, we conducted corresponding detection and observation after injection of TAA. See supplementary materials (Supplementary Fig.3) for detailed results.

2- why the authors used the elevated plus maze in the context of assessing locomotor function? The EPM is more specific test to anxiety state assessment rather than locomotion, moreover, by checking the results, no data is provided with relevance to the EPM test!

[Response]

Thank you for your careful review and constructive suggestions regarding our manuscript. I apologize for the expression in this article, because there are too many versions of the writing method, which leading to some misuse. As you said, the EPM is indeed the evaluation of emotion, we did not carry out the relevant detection, and removed the method. I am very sorry for this.

3-No details are provided in regarding the static and dynamic parameters to be assessed using the Cat Walk test.

[Response]

The total length of the Cat Walk test used in our experiment was 130 cm, and the Walkway length and Walkway Width for detecting the movement of mice were 23.86 cm and 4.08 cm.

4-no indication regarding the volume and the administration route (i.p or s.c) of the dextrose, KCl and NaCl solution; the volume of the administered solution depends on the injection route.

[Response]

The volume we injected was 25 mL/kg. We mainly refer to the previous literature^[1] and the previous research program of our research group^[2-3].

Reference:

- [1] Farjam M, Dehdab P, Abbassnia F, et al. Thioacetamide-induced acute hepatic encephalopathy in rat: behavioral, biochemical and histological changes. *Iran Red Crescent Med J.* 2012; 14(3):164-70.
- [2] Yang YL, Li JJ, Ji R, et al. Abnormal chloride homeostasis in the substantia nigra pars reticulata contributes to locomotor deficiency in a model of acute liver injury. *PLoS One.* 2013; 8(5): e65194.
- [3] Bai Y, Bai Y, Wang S, et al. Targeted upregulation of uncoupling protein 2 within the basal ganglia output structure ameliorates dyskinesia after severe liver failure. *Free Radic Biol Med.* 2018; 20 (124):40-50.

5-Remove “material and method” from page 8

[Response]

Thank you for your careful review. We have deleted this part.

Under results section:

the authors should provide an additional photomicrograph of a coronal section through the targeted brain area showing the site of intracerebral injection.

[Response]

The virus injection needle and targeted brain area was also shown in Supplementary Fig. 5.

2-The HE stained liver slices showed in fig 1, 5 and 7 should be well presented to highlight the histological lesions (high magnification)

[Response]

Thank you for your comments. We have added high magnification field to all H&E dyed pictures.

3-Captions for the last 6 figures (fig 7, 8, 9, 10, 11 and 12) are missing along with their titles.

[Response]

We have added the corresponding titles.

2-some of the presented parameters of Open field test in the results section are not mentioned in the material and methods sections (e.g main speed).

[Response]

Thank you for your comments. I'm really sorry about that. We added the corresponding open field test parameters total distance, speed, activity number and activity time in the method section.

3-As well, the cited OF parameters (the distance traveled to the center, and the number of times the animal entered the center) are not shown in the results section.

[Response]

I apologize for the inconsistency in my statement. In behavioral experiments such as open field tests, we mainly evaluated the motor ability of mice, so we only conducted statistical analysis on the total distance, speed and number of activities. And amend it.

4-In Fig1 and Fig 3, the titles should be shortened.

[Response]

Response: Thank you for your comments. We have shortened the titles, specifically as follows: Fig.1 “TAA-treated mice exhibited MHE associated liver and behavioral changes” and Fig.3 “Causal effects of SNr^{Gad2} neuron activity on motor behavior in MHE (now Fig.6)”.

Response to Reviewer #4

Reviewer #4 (Remarks to the Author):

In the current manuscript by Bai et.al., authors try to explore the molecular mechanism by which increased GABAergic tone in SNr causes bradykinesia in hepatic encephalopathy. The authors used the DREADD and TRAP2 strategies to determine that oxidative stress and mitochondrial abnormalities are involved in HE. The manuscript also shows that Gad2 expressing GABA neurons within the SNr are important contributors to locomotor injury in the HE situation. Overall, the current manuscript authors used multiple approaches to address the role of GABAergic tone in SNr in HE. Below are some of the queries I feel need to be addressed.

Major:

Major 1:

The authors took advantage of the DREADD system to explore the molecular mechanism. The authors looked at UCP2 expression upon Gi and Gq DREADD activation. However, authors should also look at the expression profile of other autophagy and oxidative stress genes upon DREADD activation or inhibition to rule out the possibility of any other mechanisms.

[Response]

Thank you for your comments.

We have performed Western blot and detected the expression levels of antioxidative stress markers of SOD1 and GPx1, as well as the autophagy markers of LC3B and Pink1, in SNr. The results from three groups including control (mCherry + CNO), chemogenetic inhibition (Gi + CNO), and chemogenetic activation (Gq + CNO) (n = 3) have shown that either chemical inhibition or chemical activation of SNr^{Gad2} neurons by DREADDs strategy would not affect the decreased levels of SOD1 and GPx1, and increased levels of LC3B and Pink1 in TAA-induced MHE mice.

We have added the contents as supplementary Fig. 6 in Supplementary file. Please check it.

Major 2:

Earlier reports showed that hyperammonemia induces activation of astrocytes and microglia, so authors should discuss the possibility of cross-talk between the neurons

and other cell types.

[Response]

Thank you for your careful review and constructive suggestions regarding our manuscript.

It has been shown that HE in patients with liver cirrhosis is triggered by cerebral edema and cerebral oxidative / nitrosative stress which bring about a number of functionally relevant alterations. These alterations are suggested to impair astrocyte / neuronal functions. We infer the beneficial effects of systemic treatment of Mito-Q and SNr-targeted overexpression of Ucp2 on oxidative stress and behavioral motor impairments might attribute to functional restore or remodeling of astrocyte and neuronal communication.

We have added the discussion in the section of Discussion. Please check it on Page 13.

Major 3:

The authors used 2 mg/kg of CNO, and earlier studies mostly used 1 mg/kg or less, as CNO can be metabolized and can have off-target effects. To rule out this possibility, authors should consider using DCZ in at least one or two major experiments to confirm the same phenotype observed.

[Response]

Thank you for your comments.

Considering the fact of off-target phenomena of CNO, we have performed associated experiments by DCZ. DCZ is a metabolic product of CNO and has a stronger effect than that of CNO. The results have shown that chemogenetic inhibition (Gi) or chemogenetic activation (Gq) of SNr^{Gad2} expressing GABA population by DCZ could ameliorate or deteriorate TAA-induced MHE, respectively. We have provided the representative confocal images, the schematic traces of the open field tests, and the statistical results of the total distance, average speed, activity numbers, and activity time by open field test, and duration by rotarod test.

We have added the contents as supplementary Fig. 5 in Supplementary file. Please check it.

Major 4:

The authors should use some pharmacological interventions to validate Fig. 6. In addition to ammonia toxicity, authors need to at least discuss the possibility of neuronal

crosstalk between the CNS and peripheral tissues, which may also contribute to the claimed observations.

[Response]

Thank you for your comments.

In the present study, we have used Mito-Q as a mitochondria-targeted antioxidant drug. Mito-Q has been shown to target to accumulate in the matrix facing the surface of the mitochondrial inner membrane where it is optimally positioned to reduce mitochondrial-resourced ROS. Mito-Q is important for central nervous system because it could rapidly cross through the blood-brain barrier and accumulate in the brain. In fact, Mito-Q has been used safely in Phase II clinical trials for Parkinson's disease patients and severe liver injury patients. In the present research, we have revealed that Mito-Q supplementation could improve liver damage, metabolism impairment, and mitochondrial dysfunction induced by TAA or BDL. Moreover, the targeted overexpression of Ucp2 has further shown the beneficial effect on the rescue of liver damage, metabolism impairment, and mitochondrial dysfunction induced by TAA. These results have indicated the key role of the communication of the central nervous system and peripheral tissues in different types of encephalopathy resulting from the injuries of liver, gut, kidney, and so on. The remission of the injuries of peripheral tissues could promote the recovery of neuronal function in the brain, at the same time, neuronal homeostasis contributes to the functional maintenance of peripheral tissues. We should pay more attention to the reciprocal effect of neurons and hepatic cells in the clinical treatment of HE.

We have added the contents into the section of Discussion.

Please check it at the third paragraph.

Minor:

Minor 1:

The current study definitely helps us understand the role of SNr^{Gad2} expressing neurons in HE, but claiming a new target is far-fetched. Maybe the language can be toned down.

[Response]

Thank you for your advice.

According to your comments, we have modified all sentences with such tone. For example, we have revised the last sentence of the original manuscript of “This

conclusion provided new targets for HE therapy.” to “Our discovery provides a new perspective into the relationship between mitochondrial and HE, and potential therapeutic target for HE.” in the present manuscript.

Minor 2:

It would also be nice if the authors considered describing a few sentences about the limitations of the study.

[Response]

Thank you for your advice.

There are three limitations here. First, we did not perform the totally same experiments on BDL-induced MHE model as that in TAA mice. Second, we did not study the involvement of excitatory neuronal population and glial cells in HE pathology. Third, we were unclear the detailed changes of the downstream of the activated SNR^{Gad2} neurons in MHE.

We have added the contents into the section of Discussion.

Please check it at the third paragraph.

Minor 3:

UCP2 KD seems to be around 50–60%, and this KD is achieving complete loss of phenotype. Can the authors confirm the percentage of KD they observed?

[Response]

The results in this part represent the percentage of UCP2 knock-down and overexpression, which are both reflected in the original data. In order to make readers more clearly understand the percentage reduction in this part, we have added a value in the result. Western Blot results showed there was a significant increase (56.36%) or decrease (50.29%) of Ucp2 levels after Ucp2OE or Ucp2KD treatment, respectively (Fig. 7B).

Minor 4:

In the second line of the introduction, I believe it is acute or chronic liver failure.

[Response]

Thank you for your comments.

It is our mistake.

There is indeed acute or chronic liver injury (ALF or CLF).

I apologize for that.

Minor 5:

In the methods section, "Determination of ammonia levels in blood and brain," the authors talked about only blood; I need some clarification on this. Also used term killed, I'm not sure this is an appropriate term to use.

[Response]

Thank you for your comments.

It is our mistake.

We have corrected the description of "*The mice were anesthetized, and the blood sample was obtained by removing the eyeball and sacrificed. Blood was collected and centrifuged at 3500 g at 4 °C for 10 min before storage at -80 °C for testing. Blood ammonia was detected using a commercial kit (Mlbio, China). The steps were strictly performed following the instructions of the kits*" in the original manuscript to the "*The mice were anesthetized, and the blood samples were obtained by removing the eyeball and sacrificed for determination of ammonia levels in blood. The whole brains were isolated and homogenized for determination of ammonia levels in brain. Blood samples and brain samples were centrifuged at 3500 g at 4 °C for 10 min and detected using a commercial kit (Mlbio, China). The steps were strictly performed following the instructions of the kits.*" in the present manuscript.

In addition, we have corrected "killed" to "sacrificed".

Thank you for your comments.

REVIEWERS' COMMENTS

Reviewer #1 (Remarks to the Author):

Thank you for addressing my points.

Reviewer #2 (Remarks to the Author):

I appreciate the efforts made by the authors to address most of my concerns and questions in the first review. The authors have done a substantial amount of additional experiments to strength their arguments, esp the additional measurements of various metabolic indicators in their Mito-Q treated animals which showed some brilliant improvement. All the experimental results are well presented that warrant consideration of publication.

Despite the authors mentioned in the rebuttal letter that this revised manuscript was proofread by a professional editor company, there are still lots of grammatical mistakes and typos which made the manuscript difficult to read and understand. This revised manuscript must be seriously proofread again before publishing in this highly reputed international journal.

Thanks again for this manuscript.

Reviewer #3 (Remarks to the Author):

1-Please check again the supp fig 5, no image was provided of the micrograph coronal section through the targeted brain area showing the site of intracerebral injection.

2-The authors stated in their manuscript, under Material and methods section (lines 464-465) "Both TAA model and BDL model were used as minimal hepatic encephalopathy (MHE) models in the present study" while according to the adopted protocol as stated in lines 468-469 "TAA (150 mg per kg of body weight) were administered once daily for consecutive 3 days", There still a confusion on the used TAA model; When TAA is administrated at a dose of 150 mg/kg (3 injections), it is for inducing acute HE and not minimal HE.

Additionally, in the response letter you mentioned the following "We have the following four reasons to use TAA mice model in the present study Second, there is a literature review to indicate that TAA induced symptoms in mice are similar to that of human acute hepatic liver disease"

3-the injected volume of Dextrose, Kcl, NaCl (25ml/kg) should be provided in the M and M section.

El Hiba Omar

Reviewer #4 (Remarks to the Author):

I am happy with the responses.

Response to Reviewer #1

REVIEWER COMMENTS

Reviewer #1 (Remarks to the Author):

Thank you for addressing my points.

[Response]

Dear reviewer, thank you so much for your constructive suggestions regarding our manuscript. It is your guidelines which makes the manuscript complete and clear.

Response to Reviewer #2

Reviewer #2 (Remarks to the Author):

Point 1:

I appreciate the efforts made by the authors to address most of my concerns and questions in the first review. The authors have done a substantial amount of additional experiments to strength their arguments, esp the additional measurements of various metabolic indicators in their Mito-Q treated animals which showed some brilliant improvement. All the experimental results are well presented that warrant consideration of publication.

[Response]

Dear reviewer, thank you so much for your constructive suggestions regarding our manuscript. It is your guidelines which makes the manuscript complete and clear.

Point 2:

Despite the authors mentioned in the rebuttal letter that this revised manuscript was proofread by a professional editor company, there are still lots of grammatical mistakes and typos which made the manuscript difficult to read and understand. This revised manuscript must be seriously proofread again before publishing in this highly reputed international journal.

[Response]

Thank you so much for your comments.

We express our appreciation to Yi-han Zhao, translator currently working for Xi'an Aeronautic Polytechnic Institute, to review our manuscript. All changes have been marked in red.

We have revised the title to "*Effects of oxidative stress on hepatic encephalopathy*"

pathogenesis in mice".

We have revised the abstract to "*Oxidative stress plays a crucial role in the pathogenesis of hepatic encephalopathy (HE), but the mechanism is not clear. GABAergic neurons in substantia nigra pars reticulata (SNr) contribute to the motor deficit of HE. The present study aims to investigate the effects of oxidative stress on HE in male mice. The results confirm the presence of oxidative stress in both liver and SNr in two types of HE models of thioacetamide (TAA) and bile duct ligation (BDL) in mice. Systemic mitochondria-targeted antioxidative drug mitoquinone (Mito-Q) rescues mitochondrial dysfunction as well as oxidative injury in SNr, so as to restore the locomotor impairment in TAA and BDL mice. Furthermore, the GAD2-expressing SNr population (SNr^{GAD2}) is activated by HE. Both SNr^{GAD2}-targeted overexpression of mitochondrial uncoupling protein 2 (UCP2) and SNr^{GAD2}-targeted chemogenetic inhibition rescue mitochondrial dysfunction in male TAA-induced HE. These results define the key role of oxidative stress in HE pathogenesis.*"

We have deleted the sentence "*The present results propose reliability of mitochondrial redox stress - based treatment in clinical hepatic encephalopathy at the early stage*" in the abstract to tone down claims of clinical treatment.

We have deleted the word "first" in: P3, Line 47 and Line 64; P4, Line 80 and Line 90; P6, Line 159; P17, Line 471; P18, Line 494; P24, Line 651.

We have deleted the word "new" in: P4, Line 78; P5, Line 131; P15, Line 411.

We have not used the words of novel / extremely / outstanding.

We have replaced the word "Gad2" by the word "GAD2". Please check: P2, Lines 29 - 32; P3, Line 59; P5, lines: 106, 107; P6, lines 131, 153, 169, 170, 176-178; P7, lines: 190, 204, 234, 235, 237 - 240, 242; P11, lines: 275, 281; P13, lines: 338, 340, 343; P14, line 370; P15, lines: 380, 399, 401; P16, lines: 407, 411, 414, 422; P18, line: 495; P21, lines: 556, 558, 566, 576; P33, line 899; P38, lines: 930, 931; P39, lines: 932, 937; P40, lines: 946-948; P41, line 957; P42, lines: 970-974.

We have replaced the word "Ucp" by the word "UCP". Please check: P2, Lines: 32, 35; P5, Lines: 97, 103, 119; P6, line 148; P9, line 218; P9, lines: 221, 222, 228, 230, 233 - 237; P10, lines: 244, 245, 247, 264; P11, line 296; P12, line 326; P14, line 372; P32, lines: 871, 872, 879; P33, line 901; P36, line 916.

We have replaced the word "Fos" by the word "FOS". Please check: P4, Line 70; P7, lines: 156, 164, 173, 183; P12, line 358; P15, lines: 379, 388, 393; P20, lines: 547, 552, 553.

We have replaced the word “GPx” by the word “GPX”. Please check: P4, Line 87; P5, lines 104 and 113; P6, line 125; P9, lines: 219, 223; P12, lines: 306; 308; P31, line 862; P32, lines: 871, 879; P33, line 901; P36, line 916.

We have replaced the word ““Pink” by the word “PINK”. Please check: P5, Lines: 95, 112; P6, line 148; P9, lines: 219, 224; P32, lines: 870, 881; P35, line 902; P36, line 917.

We have replaced the word “Drp” by the word “DRP”. Please check: P5, Lines: 98, 103; P6, line 148; P32, lines: 874, 882, 883; P35, line 903; P36, line 919; P43, line 980.

We have replaced the word ““Mff” by the word “MFF”. Please check: P5, Lines: 99, 100; P6, line 148; P32, lines: 874, 883; P35, line 903; P36, line 919; P43, line 980.

We have replaced the word ““Fis” by the word “FIS”. Please check: P5, Lines: 100, 103; P6, line 148; P32, lines: 874, 883; P35, line 903; P36, line 919; P43, line 980.

We have replaced the word “Mfn” by the word “MFN”. Please check: P5, Line 101; P6, line 149; P32, lines: 874, 883; P35, line 903; P36, line 919; P43, line 980.

We have checked and revised the use of italics, bold font, underlining or speech marks/quotation marks. All changes are marked in red.

We have revised the legends of all main figures and supplementary figures. All changes are marked in red.

Point 3:

Thanks again for this manuscript.

[Response]

Thank you so much!

Response to Reviewer #3

Reviewer #3 (Remarks to the Author):

Dear Dr. El Hiba Omarreviewer,

Thank you so much for your constructive suggestions regarding our manuscript.

Point 1:

1-Please check again the supp fig 5, no image was provided of the micrograph coronal section through the targeted brain area showing the site of intracerebral injection.

[Response]

Thank you for comment.

We have added a schematic drawing showing the site of intracerebral injection, which is Supplementary Fig. 5e.

Point 2:

2-The authors stated in their manuscript, under Material and methods section (lines 464-465) “ Both TAA model and BDL model were used as minimal hepatic encephalopathy (MHE) models in the present study” while according to the adopted protocol as stated in lines 468-469 “TAA (150 mg per kg of body weight) were administered once daily for consecutive 3 days”, There still a confusion on the used TAA model; When TAA is administrated at a dose of 150 mg/kg (3 injections), it is for inducing acute HE and not minimal HE.

Additionally, in the response letter you mentioned the following “We have the following four reasons to use TAA mice model in the present study Second, there is a literature review to indicate that TAA induced symptoms in mice are similar to that of human acute hepatic liver disease”

[Response]

Thank you for comment.

Hepatic encephalopathy (HE), is a debilitating neurological complication of liver disease/failure characterized by cognitive, psychiatric and motor disturbances. According to 2021 ISHEN guidelines on animal models of HE ^[1], three types of HE, A ~ C, have been defined based on degree and severity of liver failure, disease or impairment (including degree of portal-systemic shunting). Among them, TAA model

is Type A HE, associated with acute liver failure resulting from the rapid onset of hepatocyte necrosis and severe inflammation, and a model induced chronic liver injury (BDL) is Type C HE, associated with chronic liver disease. A common feature of TAA and BDL HE models is hyperammonemia. In the present study, we use TAA and BDL HE models.

In addition, as described in the review by Felipo V [2], based on the severity of clinical manifestation, HE is graded as 4 grades of 1 ~ 4. Patients without evident HE symptoms may present with minimal HE, which is characterized by mild cognitive impairment and bradykinesia. According to the report from Sun X et al. [2], the minimal HE model is set up by repetitive administrations of TAA. In the present study, we also used such TAA model.

Given that there is no consensus reached on the issue that whether TAA model is seen as minimal HE, we agree to careful use of the term “minimal HE” in the present study. So we delete “minimal” from the title, abstract and the associated figures legends to avoid ambiguity.

References:

- [1] DeMorrow S, Cudalbu C, Davies N, Jayakumar AR, Rose CF. 2021 ISHEN guidelines on animal models of hepatic encephalopathy. *Liver Int.* 2021; 00: 1-15.
- [2] Felipo V. Hepatic encephalopathy: effects of liver failure on brain function. *Nat Rev Neurosci.* 2013; 14(12): 851-858.
- [3] Sun X, Han R, Cheng T, Zheng Y, Xiao J, So KF, Zhang L. Corticosterone-mediated microglia activation affects dendritic spine plasticity and motor learning functions in minimal hepatic encephalopathy. *Brain Behav Immun.* 2019; 82: 178-187.

Point 3:

3-the injected volume of Dextrose, Kcl, NaCl (25ml/kg) should be provided in the M and M section.

[Response]

Thank you for comment.

We have revised this sentence from original “*Subcutaneous injection of 25 mL/kg 0.45% NaCl, 5% dextrose, and 20 mM KCl was administered 12 h after TAA injection for the first time to prevent hypoglycemia and electrolyte imbalance.*” to the present “*Subcutaneous injection of 25 mL/kg 77.0 mM NaCl, 227.5 mM dextrose, and 20.0 mM KCl was administered 12 h after TAA injection for the first time to prevent hypoglycemia*”

and electrolyte imbalance.”.

Please check Page 16, lines 429 and 430.

Response to Reviewer #4

Reviewer #4 (Remarks to the Author):

I am happy with the responses.

[Response]

Dear reviewer, thank you so much for your constructive suggestions regarding our manuscript. It is your guidelines which makes the manuscript complete and clear.